

# Software process improvement: a systematic mapping study on the state of the art

Marco Kuhrmann[1], Philipp Diebold[2] and Jürgen Münch[3,4]

[1] Maersk Mc-Kinney Moller Institute—Software Engineering, University of Southern Denmark, Odense, Denmark
[2] Process Engineering, Fraunhofer Institute for Experimental Software Engineering, Kaiserslautern, Germany
[3] Herman Hollerith Center, Böblingen & Reutlingen University, Böblingen, Germany
[4] Department of Computer Science, University of Helsinki, Helsinki, Finnland

Corresponding author
Marco Kuhrmann,
kuhrmann@mmmi.sdu.dk

## ABSTRACT

Software process improvement (SPI) has been around for decades: frameworks are proposed, success factors are studied, and experiences have been reported. However, the sheer mass of concepts, approaches, and standards published over the years overwhelms practitioners as well as researchers. What is out there? Are there new trends and emerging approaches? What are open issues? Still, we struggle to answer these questions about the current state of SPI and related research. In this article, we present results from an updated systematic mapping study to shed light on the field of SPI, to develop a big picture of the state of the art, and to draw conclusions for future research directions. An analysis of 769 publications draws a big picture of SPI-related research of the past quarter-century. Our study shows a high number of solution proposals, experience reports, and secondary studies, but only few theories and models on SPI in general. In particular, standard SPI models like CMMI and ISO/IEC 15,504 are analyzed, enhanced, and evaluated for applicability in practice, but these standards are also critically discussed, e.g., from the perspective of SPI in small-to-medium-sized companies, which leads to new specialized frameworks. New and specialized frameworks account for the majority of the contributions found (approx. 38%). Furthermore, we find a growing interest in success factors (approx. 16%) to aid companies in conducting SPI and in adapting agile principles and practices for SPI (approx. 10%). Beyond these specific topics, the study results also show an increasing interest into secondary studies with the purpose of aggregating and structuring SPI-related knowledge. Finally, the present study helps directing future research by identifying under-researched topics awaiting further investigation.

# INTRODUCTION

Software process improvement (SPI; according to *Humphrey, 1989*) aims to improve software processes and comprises a variety of tasks, such as scoping, assessment, design and realization, and continuous improvement, e.g., *Münch et al. (2012)*. In this field, a number of SPI models competes for the companies' favor, success factors to support

SPI implementation at the large scale and the small scale are studied, and a multitude of publications report on experiences in academia and practice. *Horvat, Rozman & Györkös (2000)* consider SPI an important topic (regardless of the company size), as many companies put emphasis on the software process and its adaptation to the company context (*Diebold et al., 2015*; *Vijayasarathy & Butler, 2015*; *Theocharis et al., 2015*) to address different improvement goals, such accelerating software development or improving software quality.

However, SPI is a diverse field: on the one hand, a number of standards is available, e.g., the Capability Maturity Model Integration (CMMI) or ISO/IEC 15504. On the other hand, these standards are criticized oftentimes, as for instance by *Brodman & Johnson (1994)*, *Staples et al. (2007)* and *Coleman & O'Connor (2008)*. Dictating processes and/or process improvement programs can lead to serious organizational "immune reactions" (*Baddoo & Hall, 2003*), e.g., of developers (*Umarji & Seaman, 2008*) and entire companies due to lacking resources (*Hall, Rainer & Baddoo, 2002*). In response, several tailored standard SPI models or custom SPI approaches are proposed, inter alia, to better address needs of small and very small companies, e.g., *Raninen et al. (2012)*, *Rozman et al. (1997)* and *Pino et al. (2009)*, or to adapt agile principles in the improvement process (*Salo & Abrahamsson, 2007*). Moreover, since SPI is mainly a human endeavor, much research was spent to study human factors, e.g., *Stelzer & Mellis (1998)*, *Allison (2010)*, *Viana et al. (2012)* and *Laporte & O'Connor (2014)*. Those factors, furthermore, play an important role when SPI is conducted at the global scale, as for instance described by *Paulish & Carleton (1994)*, or if large companies want to deploy agile processes as for instance presented by *Hannay & Benestad (2010)* or *Korhonen (2013)*. Beyond, we find numerous experience reports, guidelines, and tools—all together providing a huge body of knowledge on SPI. However, despite this comprehensive body of knowledge, from the authors' perspective, we lack a big picture of SPI and we still struggle to answer questions like: What is out there? What are open issues? Are there new trends and emerging approaches, and if yes, what are the new trends? What is the current state of SPI and related research after all?

**Problem statement & objective.** The field of SPI evolved for decades and provides a vast amount of publications addressing a huge variety of topics. Still, we see new method proposals, research on success factors, and plenty of experience reports. Yet, missing is a big picture that illustrates where SPI gained a certain level of saturation, what are the hot topics, and what are unresolved issues calling for more investigation? To better understand the state of the art in SPI, we aim to analyze the whole publication flora to draw a big picture on SPI. Our overall goal is *not* to judge particular SPI research directions, but to provide the focus points of the past and to illustrate emerging/unresolved areas to show the directions for future research in this field.

**Contribution.** In this article, we present findings from an updated comprehensive systematic mapping study. Starting with a curiosity-driven study, in two stages, we conducted a broadband search in six literature databases and one meta-search engine to harvest SPI-related publications from the past 26 years, and we incrementally analyzed the resulting 769 publications for publication frequency, research type facet, contribution type facet, and we categorized the found publications using a set of 40 metadata attributes.

We draw a big picture showing that the majority of the publications on SPI either proposes custom/new approaches (i.e., models or frameworks) or is of philosophical nature (i.e., collecting, structuring, and analyzing knowledge). Our results show a constant publication of new approaches while evaluation of these proposals is scarcely available. Our data shows rare evidence and, notably, missing long-term and independently conducted replication studies. However, the data also reveals some (still) emerging topics, e.g., SPI for very small and medium-sized companies, and SPI in the context of lean and agile methods.

**Context & previously published material.** The present study is a substantial update of our initial study published in *Kuhrmann et al. (2015)*. In the course of updating the study, in particular, we added the following procedures/content: to provide an instrument that allows for continuously updating the study, we defined a new data collection procedure (Appendix 'Data collection in the study update'), which we implemented to carry out the update presented here. The update adds 141 new papers to the result set, which now contains 769 papes in total. Furthermore, we modified the data classification approach. To achieve higher precision, we defined 40 metadata attributes, and we applied these attributes to the dataset while excluding the focus type facet from the analysis (cf. 'Threats to validity'). Finally, while our initial study aimed to identify major trends, in this article, we provide a more detailed analysis of the trends found using the new classification.

**Outline.** The remainder of this article is organized as follows: 'Related Work' summarizes and discusses related work. In 'Research Design,' we detail the study's overall research design. Since this article presents an updated systematic mapping study, the article's appendix details the original and updated research methods as well as required reference data. We present and discuss the study results in 'Study Results and Discussion', and conclude the article in 'Conclusion & Future Work.'

## RELATED WORK

Literature on Software Process Improvement is rich and addresses a variety of topics. Yet, available secondary studies mainly focus on investigating success factors, e.g., *Monteiro & De Oliveira (2011)*, *Bayona-Oré et al. (2014)* and *Dybå (2000)*. Some studies provide insights into selected SPI topics, as for instance: *Helgesson, Höst & Weyns (2012)* review maturity models, and *Hull et al. (2002)* and *El-Emam & Goldenson (2000)* review different assessment models. *Pino, García & Piattini (2008)* contribute a review on SPI in the context of small and very small companies, and *Staples & Niazi (2008)* study motivating factors to adopt CMMI for improvement programs, while *Müller, Mathiassen & Balshøj (2010)* study SPI in general from the perspective of organizational change. All these representatively selected studies address specific topics, yet they do not contribute to a more general perspective on SPI. Such general studies are scarcely to find. For instance, *Rainer & Hall (2001)* analyze some 'core' studies on SPI for the purpose to work out addressed topics and gaps in the domain. However, they select few studies of which they assume to be good representatives thus providing a limited picture only. In terms of analyzing the entire domain and providing new (generalizable) knowledge, *Unterkalmsteiner et al. (2012)* contribute a systematic review on the state of the art of evaluation and measurement in

SPI. They conduct a systematic review for the purpose of synthesizing a list of evaluation and measurement approaches, which they also analyze for the practical application.

The study at hand does not aim at generating generalizable knowledge for one or more SPI-related topics in the first place. The purpose of the present study is to draw a big picture of the current state of the art of SPI in general. That is, as there is no comparable study available, this article closes a gap in literature by providing a comprehensive picture of the development of the field of SPI over time and by summarizing the current state of the art. Other than, e.g., *Rainer & Hall (2001)* or *Unterkalmsteiner et al. (2012)*, we use the mapping study instrument according to *Petersen et al. (2008)* as research method and to present our results. Therefore, our study does not address one specific aspect/topic, but aims to draw a general picture from a "bird's-eye perspective" to pave the way for further topic-specific and more detailed studies.

# RESEARCH DESIGN

In this section, we present the overall study design. After describing the selected research method, we introduce the research questions, and describe the different instruments used for data collection and analysis, and the validity procedures.

## Research method

In this study, we ground the overall research approach in the procedures implemented for our previously published initial study. In *Kuhrmann et al. (2015)*, we followed an approach in which we applied different methods from *systematic literature reviews* (SLR) according to *Kitchenham & Charters (2007)* and *systematic mapping studies* (SMS) as presented by *Petersen et al. (2008)*. While carrying out the study update, we used and improved the methods applied, which was necessary to develop a strategy that allows for continuous study updates. Figure 1 shows the overall research approach for which we provide details in subsequent sections.

**Initial study.** The initial study was designed as a *breadth-first search* to cover the SPI domain as complete as possible. In February 2013, we performed the study preparation, conducted a series of test runs, and refined the search queries iteratively. End of April 2013, we conducted the main search, which resulted in about 85,000 hits. As we expected this large number of results and in order to support the dataset cleaning, we defined filter questions, which we applied to the initial result set. When the initial result set was cleaned, we performed a voting procedure to select the relevant publications from the result set. Based on this selection, we developed the classification schemas (by manual sampling as well as tool-supported) and harmonized the dataset (e.g., completion of keyword lists).

**Study update procedure.** As one of the goals was to develop an instrument to provide a "heartbeat" of the whole field, having a strategy available to continuously update and refine the study was an imperative. Therefore, after having conducted and analyzed the initial study, we collected lessons learned and developed the update strategy. The outcome is shown in the right part of Fig. 1. The revised approach comprises a changed data collection procedure (Appendix 'Data collection in the study update') and an improved study

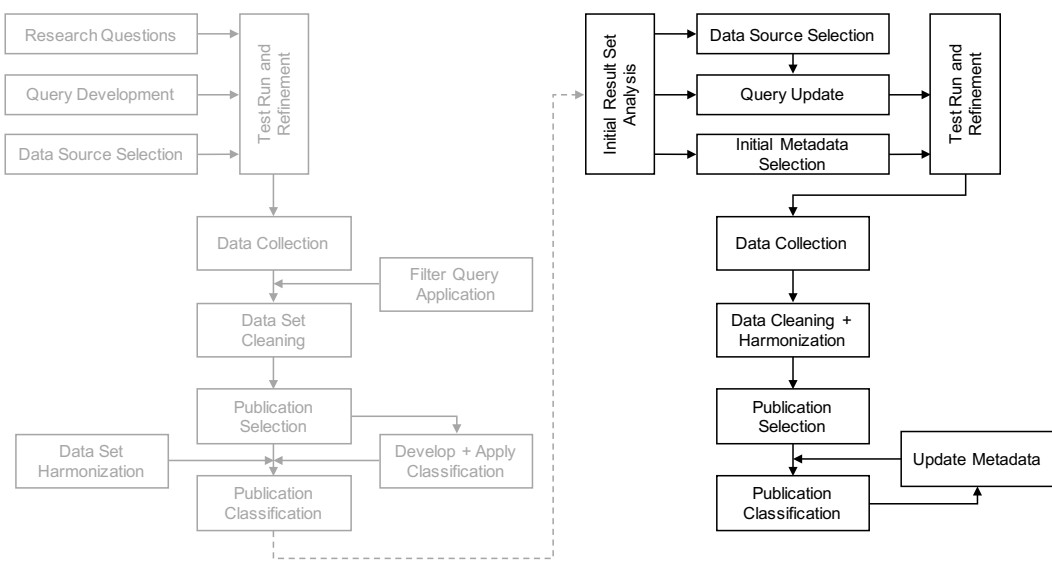

**Figure 1** Overview of the applied research methods in the initial study (left part of the figure) as well as in the study update procedure (right part of the figure).

classification procedure ('Analysis and classification'). The update procedure was defined in August 2015, and the actual update was performed from September 2015 to November 2015. In subsequent sections, we describe this new strategy, whereas the particular changes are documented in detail in the appendix of this article.

## Research questions

Our objective is to capture the domain of *Software Process Improvement* (SPI), to provide a continuously updated snapshot of the available publication pool, and to investigate research trends. Therefore, we define the following research questions:

RQ 1:     *What is the general publication population on SPI?* This research question aims to get an overview of the general publication pool on SPI. We are interested in getting information regarding publication count, frequency and, eventually, an overview of the different research type facets addressed by the found publications.

RQ 2:     *What is the contribution population?* Based on the found publications, we are interested in the addressed topics and major contributions (e.g., SPI models, theories, secondary studies, and lessons learned) to work out the SPI topics to which research contributed so far.

RQ 3:     *What trends in SPI and SPI-related research can be observed?* The third research question aims at investigating the focus points addressed by SPI research so far, and to work out gaps as well as trends. This research question shall pave the way to direct future research on SPI.

## Data collection procedures

As mentioned in 'Research method', due to lessons learned in the initial study and in order to provide a feasible strategy for study updates, the research approach had to be improved.

**Table 1  Spreadsheet layout to collect, structure, and evaluate data.**

| Information set | Attributes and description |
|---|---|
| Study Keys | Running No (unique number in the dataset), No (unique number in the database), Database |
| Content | Title, Authors, Year, Keywords/Tags, Abstract |
| Voting | Relevance (defined during further analysis and voting by the different authors, cf. 'Analysis preparation'), Disc (decision field to be set in workshops if a paper was marked for discussion), Result (paper is in or out) |
| Publication Vehicle | A publication is published in either a journal, magazine, conference, workshop, book, or miscellaneous (cf. Fig. 2) |
| Research Type Facet | Classification of a paper according to the research type facet (RTF) as proposed by *Wieringa et al. (2005)* |
| Contribution Type Facet | Classification of a paper according to the contribution type facet (CTF) according to *Shaw (2003)* (see also *Petersen et al., 2008*) |
| Metadata | Collection of metadata per paper according to the structure from Fig. 2 |
| Further Information | Further information and/or further metadata to be collected |

The most significant changes regarding the data collection procedure are described in Appendix 'Data Collection Procedures.' In the following, we describe the actual data collection procedure applied to the present study.

**Query construction.** The basic queries were already developed in the initial study (Appendix 'Query construction'). After the initial result set analysis, the query strings were critically reviewed and updated (Fig. 1). However, no new search terms were added, only the structure of the queries required some updates to address the new data source that serves as main input. In a nutshell, due to the change of the search engine, the main search strings $S_1$–$S_8$ were integrated with the context and filter queries, which were required in the initial study to help querying the different literature databases. The full new search queries can be depicted from Table 11 (Appendix 'Search queries').

**Data sources and data format.** In the present study, after reviewing the initial study designs and results, we looked for more efficient ways to fetch papers for the update and eventually opted for Scopus[1] as new search engine. Having executed the different queries, obtained data was merged into one spreadsheet that structures the data and contains the attributes shown in Table 1. The data structure shown in Table 1 follows the structure used in the initial study.

## Analysis procedures

We describe the analysis preparation as well as the steps conducted to answer the research questions.

### Analysis preparation

We performed an automated search that required us to filter and prepare the result set. The data analysis is prepared by harmonizing the data, performing a 2-staged voting process, and integrating the initial and the update data set to prepare the result set analysis.

[1] Scopus is available from: http://www.scopus.com. Before we made this decision, we tested Scopus: We took some initial search queries (Table 10), queried Scopus, and compared the obtained data with the original datasets. We then iteratively enhanced the Scopus search strings and, eventually, defined the following quality requirement for the search: Given the trends in publication frequency and classification obtained in the initial study, we expect a similar frequency and classification for the Scopes-based search (see also 'Result overview').

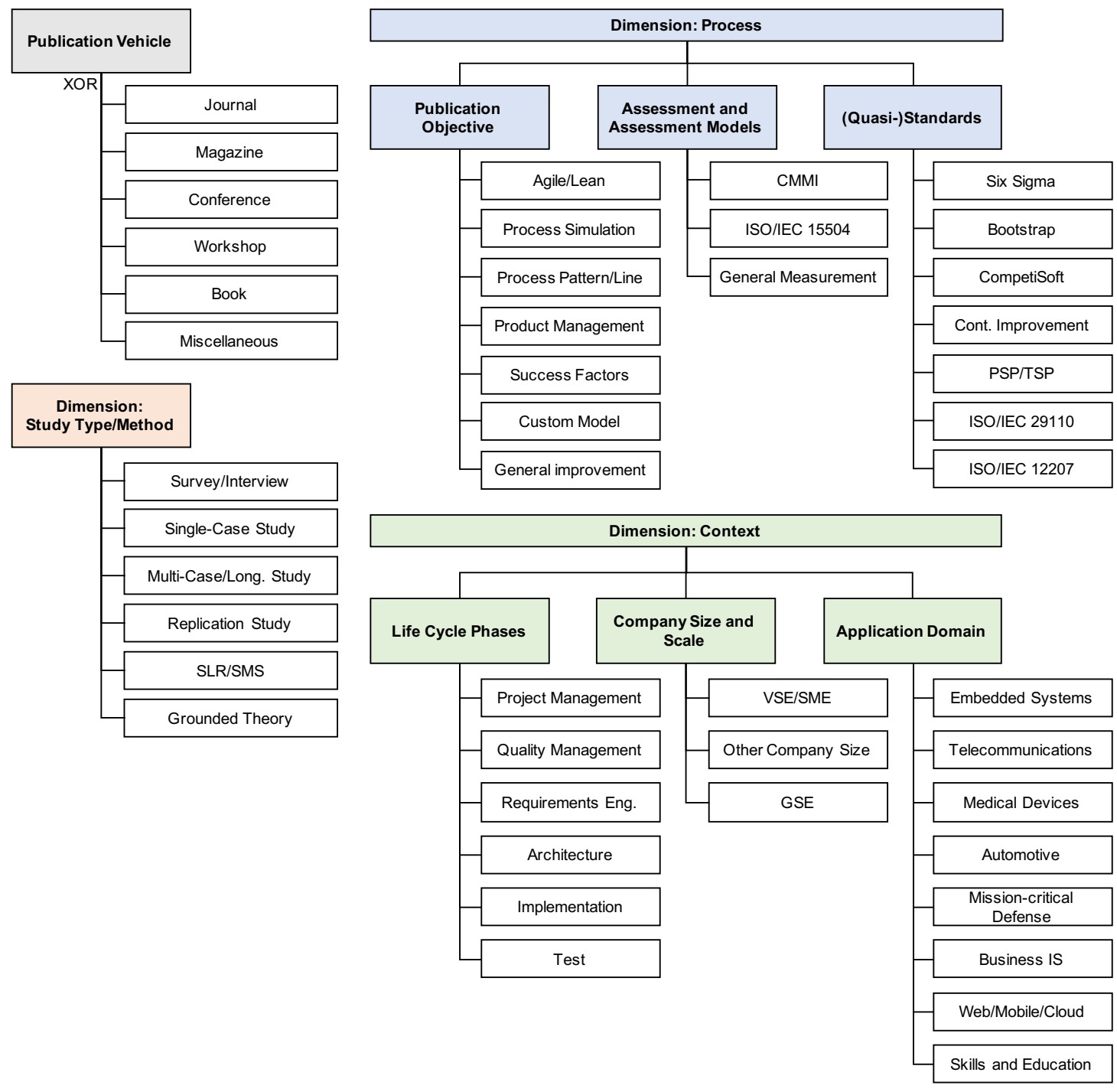

**Figure 2** Overview of the collected metadata in the study analysis phase, including publication vehicles and 40 study-specific attributes and their grouping in topic cluster (dimensions).

**Table 2** Inclusion and exclusion criteria applied to the study.

| Criteria | Description |
|---|---|
| IC$_1$ | Title, keyword list, and abstract make explicit that the paper is related to SPI. |
| IC$_2$ | Paper presents SPI-related topics, e.g., SPI models, assessments, experiences in adopting and deploying software processes, and reports on improving specific methods/practices. |
| EC$_1$ | Paper is not in English. |
| EC$_2$ | Paper is not in the field of software engineering or computer science in general. |
| EC$_3$ | Paper is a tutorial or workshop summary only. |
| EC$_4$ | Paper occurred multiple times. |
| EC$_5$ | Paper full text is not available for download. |

**Harmonization.** To make the selection of the contributions more efficient, we first integrated and cleaned the result set. We removed the duplicates, which we identified by title, year, and author list. The main instrument used was the Microsoft Excel feature to identify and remove duplicates (cf. Appendix 'Search and cleaning procedure'). This procedure was performed on the integrated result set.

**Voting.** We applied the voting procedures as described in *Kuhrmann et al. (2015)*. That is, we performed a multi-staged voting process to classify the papers as relevant or irrelevant and to build a set of publications for further investigation (Table 1, Voting). In the voting process, the inclusion and exclusion criteria listed in Table 2 guided the decision-making process. Two researchers performed individual votings (initially: publication title and abstract). If both agreed, the paper was directly included or excluded. For those papers that were not immediately agreed, workshops were performed to resolve disagreements. After the initial voting, the selection was reviewed by a third researcher for confirmation.

**Integration.** In the final step, we integrated the initial result set from *Kuhrmann et al. (2015)* with the Scopus update. Due to the expected overlaps (search year 2013), we checked the result set for duplicates again and—if necessary—removed the found duplicates.

### Analysis and classification

On the final set, the analysis and classification were performed using the abstracts and—where necessary—the complete publication. Generally, each classification step was conducted independently by two researchers, merged, discussed, and eventually checked by the third researcher. In the following, we summarize the analysis procedures used to answer our research questions.

**Research type facets.** In order to classify the publications, we rely on the classification according to the *research type facet* as proposed by *Wieringa et al. (2005)*. However, during a test classification on a small sample, we found the need to adjust the facet definitions. Table 3 lists the research type facets as applied to the result set.

**Contribution type facets.** In order to analyze what and how publications contribute to the body of knowledge, we adopted the *contribution type facets* as proposed by *Shaw (2003)*. Table 4 lists the facet types applied to the result set.

**Table 3  Applied research type facets as proposed by *Wieringa et al. (2005)*.**

| Criteria | Description |
|---|---|
| Evaluation research | Implemented in practice, evaluation of implementation conducted; requires more than just one demonstrating case study |
| Solution proposal | Solution for a problem is proposed, benefits/application is demonstrated by example, experiments, or student labs; also includes proposals complemented by one demonstrating case study for which no long-term evaluation/dissemination plan is obvious |
| Philosophical paper | New way of thinking, structuring a field in form of a taxonomy or a framework, secondary studies like SLR or SMS |
| Opinion paper | Personal opinion, not grounded in related work and research methodology |
| Experience paper | Personal experience, how are things done in practice |

**Table 4  Applied contribution type facets as proposed by *Shaw (2003)*.**

| Criteria | Description |
|---|---|
| Model | Representation of observed reality by concepts after conceptualization |
| Theory | Construct of cause–effect relationships |
| Framework | Frameworks/methods related to SPI |
| Guideline | List of advices |
| Lessons learned | Set of outcomes from obtained results |
| Advice | Recommendation (from opinion) |
| Tool | A tool to support SPI |

[2] In the initial study *Kuhrmann et al. (2015)*, the focus type facets were found inadequate for this study stage, e.g., due to variety of the topics addressed and the limitations to define proper topic clusters or the need to have multiple assignments for many papers.

**Metadata.** Instead of applying the *focus type facet*[2] to the result set, we opted for the collection of metadata. The metadata attributes of interest were initially collected and structured in a workshop in which the lessons learned from the initial study were taken into account. During the metadata collection, reviewers had the option to propose and add further attributes, i.e., the list of metadata was extended and then the result set was revisited (see also Fig. 1).

Figure 2 provides a structured overview of the metadata. In particular, we collected metadata in the following four categories: *Publication Vehicle*, *Study Type and Method*, *Process*, and *Context*. The *Publication Vehicle* is an XOR-selection, i.e., a paper is for instance either a conference paper or a journal article. The other three categories (dimensions) can comprise sub-categories and allow for multiple selection. For example, a paper can contain an SLR-based SPI model, which is confirmed using an expert interview (dimension: *Study Type and Method*), and the study can address an agile/lean custom model that adopts CMMI (dimension: *Process*) in an SME company that works in medical devices, and improves quality management and test (dimension: *Context*).

## Validity procedures

To increase the validity of our study, we implemented the following procedures: We extensively reused our initial research design, which we only modified in terms of the

**Table 5  Data collection and filtering results of the study update, and total numbers of studies after merging and cleaning initial and update datasets.**

|  | Automatic search | | | Manual selection | | Integration | |
|---|---|---|---|---|---|---|---|
|  | **Hits** | **$EC_2$** | **$EC_{1,4}$** | **Voting** | **Discussion** | **Merge** | **Final** |
| $S_1$ | 532 | 333 | 270 | 56 | 50 | | |
| $S_2$ | 4,673 | 1,402 | 880 | 74 | 71 | | |
| $S_3$ | 815 | 301 | 15 | 1 | 1 | | |
| $S_4$ | 4,223 | 1,150 | 165 | 17 | 14 | | |
| $S_5$ | 1,609 | 545 | 29 | 1 | 1 | | |
| $S_6$ | 507 | 307 | 0 | 0 | 0 | | |
| $S_7$ | 5,997 | 1,659 | 89 | 6 | 4 | | |
| $S_8$ | 330 | 227 | 2 | 0 | 0 | | |
| Total | 18,686 | 5,924 | 1,450 | 155 | **141** | 776 | **769** |

data collection procedures. Furthermore, during the whole study, we performed several quality assurance activities (partially tool-supported), iterated through the single steps, and stepwise analyzed and refined tentative result sets. During the publication selection and classification, we relied on researcher triangulation, e.g., within a rigorous multi-staged voting procedure in which two researchers carried out the initial classification and the third researcher confirmed the classification. For the development of the classification schemas, we either ground the developed schemas in external proposals or rely on flexible and extensible metadata. Finally, we continuously compared tentative results with findings from our initial study to check for general trends.

## STUDY RESULTS AND DISCUSSION

In this section, we present and discuss the results of our study. In 'Result overview', we provide an overview of the whole result set and discuss the development of the domain observed in the study update. 'RQ 1: general publication flora', 'RQ 2: result set contribution,' 'RQ 3: Trends in SPI-related Research,' answer the research questions, before we discuss our findings in 'Discussion'. Finally, we discuss threats to validity of this study in 'Threats to validity.'

### Result overview

In this section, we provide an overview of the whole result set. Since the present study is an update study, the starting point for the study at hand is the result set from *Kuhrmann et al. (2015)*. An overview of this initial result set can be taken from Table 9. The study update covers 1.5 years and comprises publications from January 2013 to July 2015. The outcomes of the search, cleaning, and merge procedures are shown in Table 5. The table shows seven papers removed in the merge procedures, which are multiple occurrences in 2013 (eight papers were found in the initial study, which were integrated with the update result set).

Figure 3 visualizes the publication frequency of the integrated result set by showing the number of publications over time including two trend lines (trend calculation basis:

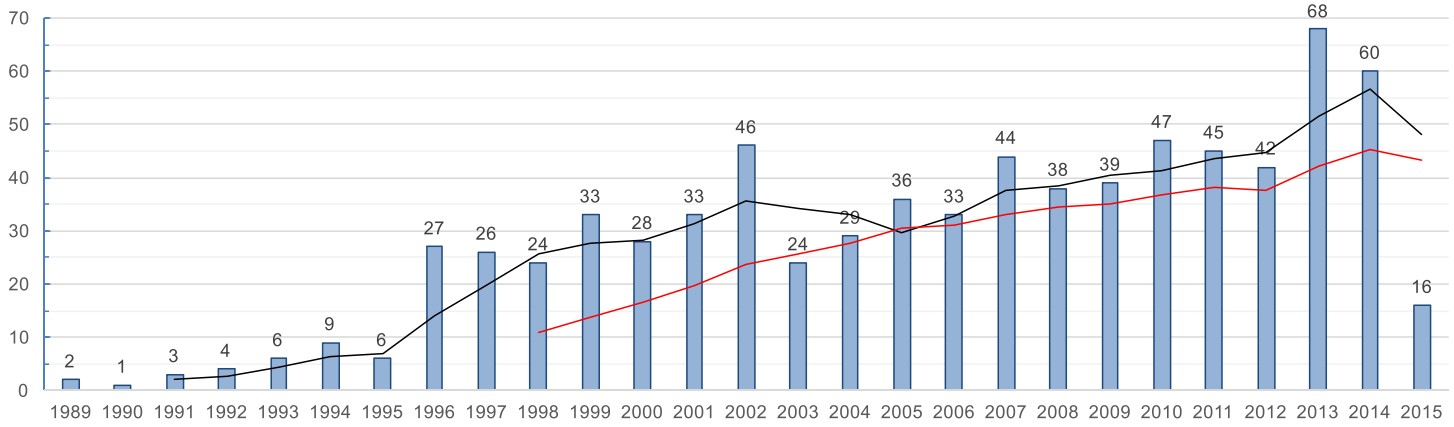

**Figure 3  Overall publication frequency (papers on SPI published per year).**

mean, 3-year and 10-year period). In 1996, the numbers show a growing interest in SPI. From this point on, SPI became an inherent part of software engineering research. Figure 3 shows periodical waves over the years starting three to five years, which is emphasized by the first 3-year trend line. Within these waves the largest gap/decrease is between 2002 and 2003. Another big jump can be seen in 2013, where the number of papers increased by approximately 50%. Furthermore, Fig. 3 shows SPI still being a field of interest, as the second 10-year trend line shows. The majority of the papers in the result set are journal articles ($n = 353$, 45.9%) and conference papers ($n = 350$, 45.5%). Magazine articles ($n = 33$) and workshop papers ($n = 30$) count for 4.3% and 3.9%. The result set does not contain books, but three papers (0.4%) that are classified as miscellaneous (mostly book chapters).

In summary, the updated study includes **769** papers on SPI published between 1989 and July 2015, which are subject to analysis. 'RQ 1: general publication flora', 'RQ 2: result set contribution,' 'RQ 3: Trends in SPI-related Research', we provide the detailed analysis to answer the research questions.

**Result set quality assurance.** As mentioned in 'Data collection procedures,' we changed the data collection procedure and, thus, we defined the quality requirement that the update result set should "harmonize" with the initial result set, i.e., the update set should show similar trends and distribution. This quality assurance was carried out using the aforementioned trend analysis and using the different research- and contribution type facets (cf. 'RQ 1: general publication flora').

Figure 4 shows the average (absolute) paper numbers and the relative distribution per category. The figure visualizes these numbers for three data points: the average in the merged dataset, and the average of the data from 1989–2012 and the study update (2013–2015), respectively. Given the trend (Fig. 3) and the about 50% increase of publications per year, still, the relative distribution of the papers in the update result set follows the general trend of the result set, which could just be observed in our initial study.

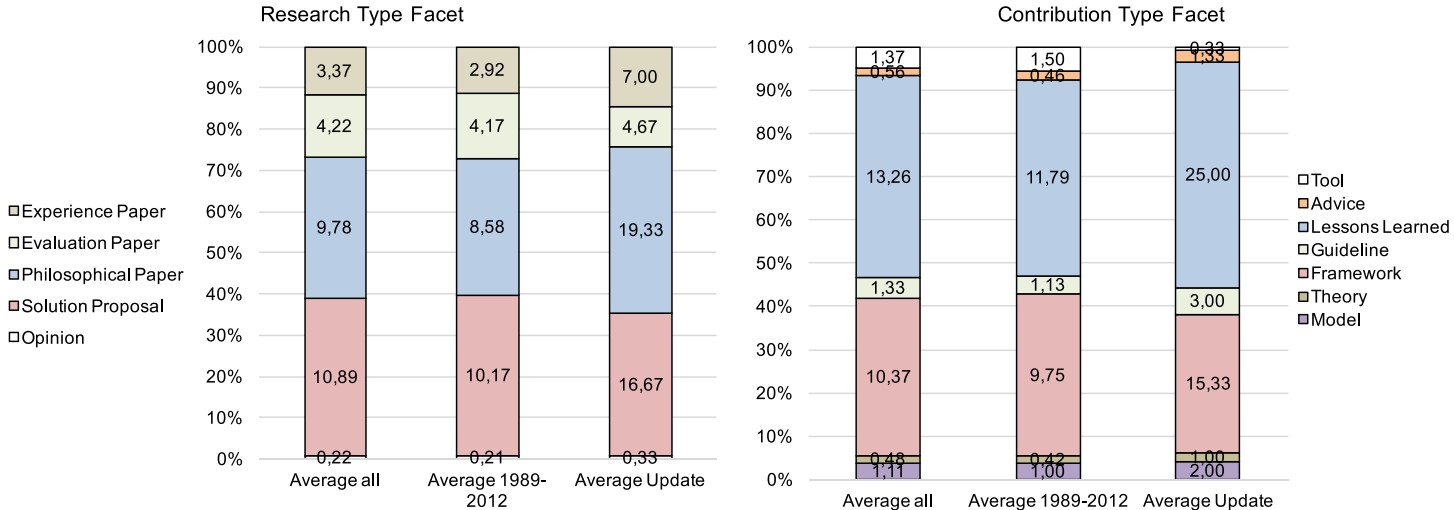

**Figure 4** **Overview of the (average) paper numbers and percentage in the result sets.** Both parts show a similar distribution in the different categories in the entire result set and the subsets addressed by the initial study and the study update.

### RQ 1: general publication flora

To get an overview of the harvested papers, we performed a categorization to define the research type facets and contribution type facets (Tables 3 and 4). To analyze the respective trends, Fig. 5 provides an integrated picture that shows the papers in the different categories and over time.

Regarding the research type facet, Fig. 5 shows a clear trend towards *solution proposals* ($n = 294$, 38.2%) and *philosophical* papers ($n = 264$, 34.3%). From the 769 papers in the result set, 114 papers (14.8%) are classified as *evaluation* papers and 91 papers (11.8%) are classified as *experience* papers. Only six out of 769 papers (0.8%) are *opinion* papers. Taking into account the general trend of the result set (Fig. 4), the classification according to the research type facet indicates a still evolving research field. Figure 5 illustrates, in average, approx. 75% of the published papers per year are either proposing "something new" or discussing an SPI-related topic from new/different perspectives, e.g., using secondary studies such as systematic reviews or mapping studies ($n = 43$, 5.6%). At the same time, only about a quarter of the published papers per year deals with evaluating research or reporting experiences.

Figure 5 (lower part) shows a similar tendency for the contribution type facet. From the 769 papers in the result set, 358 papers (46.6%) contribute lessons learned, followed by 280 papers (36.4%) that contribute custom or new frameworks. All remaining categories are below 5%, in particular, models ($n = 30$, 3.9%), theories ($n = 13$, 1.7%), guidelines ($n = 36$, 4.7%), advice ($n = 15$, 2.0%), and tools ($n = 37$, 4.8%). That is, approx. 83% of all papers either propose frameworks or discuss lessons learned, which is, again, consistent with the overall trend over time.

An impression about the progress in the field can be depicted from Fig. 6 in which we create a first systematic map relating the research- and the contribution type facet. The figure shows that most of the frameworks have to be considered a solution proposal

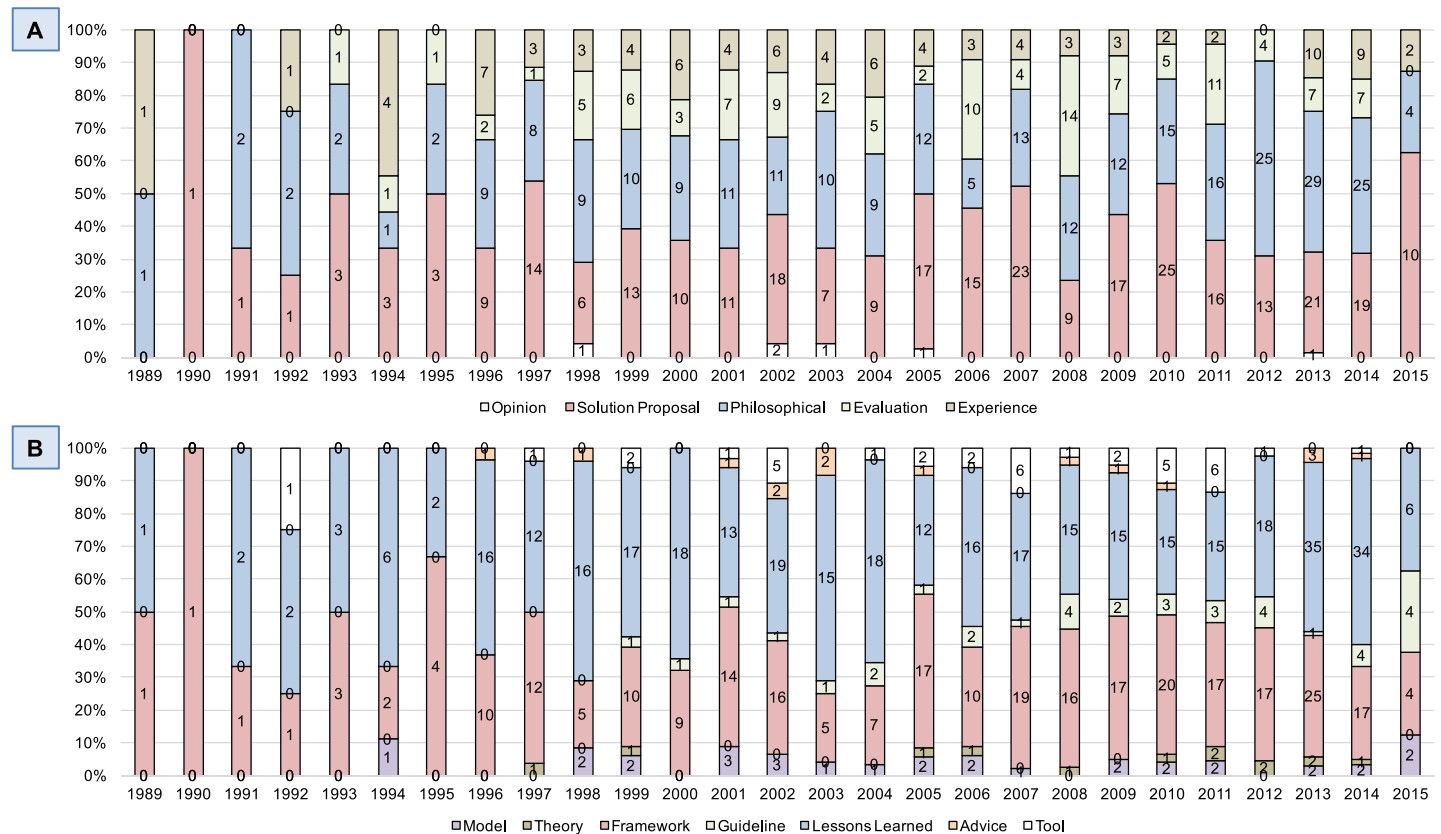

**Figure 5** Number of papers per year and relative distribution over research type facet (A) and contribution type facet (B).

(204 out of 280), but only 48 papers from the category framework are classified as evaluation research. Similar, about two third of all papers classified as lessons learned (195 out of 358) are classified as philosophical paper, i.e., lessons learned are drawn from discussion/observation in artificial or lab environments or concluded from secondary studies. From the 358 lessons-learned papers, 52 are classified as evaluation research and 81 as experience reports, which together makes approx. 37% of all lessons learned papers. Furthermore, 28 out of 30 papers that contribute models to the result set are classified as solution proposal (18 papers) or philosophical paper (10 papers). That is, models on SPI are either proposed awaiting their evaluation or those models are concluded from discussion or secondary studies, also awaiting evaluation. The same picture can be observed for theories: 11 out of 13 papers that are classified as contributing a theory are also classified as philosophical paper, and only two are classified as evaluation research.

**Summary.** From the top-level analysis using the basic classification schemas, we can observe: in the result set, we see a clear trend towards proposing new solutions, and the majority of the proposed solutions considers SPI frameworks. A second major trend is reporting lessons learned. These trends can be observed in the final result set as well as over time. Regarding the proposed frameworks, approx. 73% (204 out of 280 framework-related papers) are classified as solution proposals, i.e., method- or framework proposals without

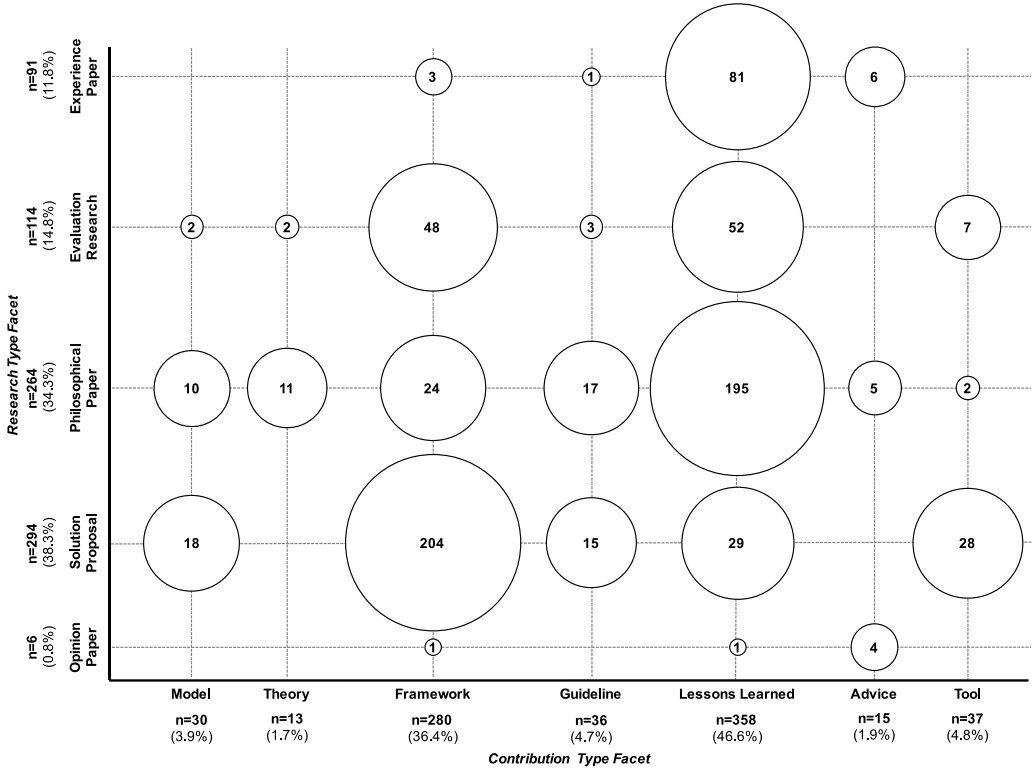

**Figure 6** Systematic map over research- and contribution type facets.

any evaluation or with theoretical or lab-based evaluation only. Similar, approx. 63% of all reported lessons learned (195 out of 358) are classified as philosophical paper, i.e., conclusions are drawn from theoretical or lab-based evaluation only. In summary, the big picture presented in this section shows a still evolving research field, which is developing new approaches and collecting lessons learned, but this field still lacks evaluated models and theories.

## RQ 2: result set contribution

In this section, we provide a more detailed perspective on the result set using the collected metadata as illustrated in Fig. 2. While classifying the result set, we collected metadata for the three dimensions *Study Type and Method*, *Process* (incl. sub-categories), and *Context* (incl. sub-categories). In addition to the publication vehicle, we defined 40 attributes, and each paper could be assigned none or many of these attributes ('Analysis and classification'). In total, for the 769 studied papers, we assigned 2,408 attribute values. All metadata assignments are summarized in Fig. 7 and discussed in the following.

**Dimension: process.** Within this dimension, we built the three categories *Assessment and Assessment Models*, *(Quasi-)Standards*, and *Publication Objective*, which provide the following insights:

Within the topic of *assessment and assessment models*, we focused on common assessment (maturity) models. Most frequently mentioned is CMMI with 170 assigned papers, followed

| Dimension | Category | Attribute | Tota | 1989 | 1990 | 1991 | 1992 | 1993 | 1994 | 1995 | 1996 | 1997 | 1998 | 1999 | 2000 | 2001 | 2002 | 2003 | 2004 | 2005 | 2006 | 2007 | 2008 | 2009 | 2010 | 2011 | 2012 | 2013 | 2014 | 2015 |
|---|---|---|---|---|---|---|---|---|---|---|---|---|---|---|---|---|---|---|---|---|---|---|---|---|---|---|---|---|---|---|
| Dimension: Process | Publication Objective | Agile/Lean | 73 | | | | | | | 1 | 1 | | 1 | | 1 | | 1 | | 6 | 1 | | 2 | 4 | 1 | 7 | 6 | 4 | 21 | 14 | 2 |
| | | Process Simulation | 23 | | | | | | 1 | | | 1 | | 4 | 2 | 2 | 4 | | | 1 | 1 | 1 | 1 | | 2 | | | | 3 | |
| | | Process Line/Patterns | 17 | | | | | 1 | | | 1 | 1 | | | | | 2 | | 2 | | 2 | | 1 | 1 | | 3 | 1 | 1 | 1 | |
| | | Product Line/Management | 9 | | | | | | | | | | | | 1 | | 2 | | | | 1 | | 1 | | | 1 | 1 | | 2 | |
| | | Success Factors | 126 | | | | | | | 1 | 5 | 1 | 3 | 3 | 5 | 6 | 10 | 5 | 3 | 5 | 2 | 6 | 6 | 6 | 12 | 7 | 12 | 13 | 13 | 2 |
| | | Custom Model | 295 | 1 | | 2 | 1 | 2 | 2 | 3 | 6 | 13 | 5 | 11 | 10 | 15 | 20 | 4 | 9 | 14 | 13 | 16 | 14 | 19 | 24 | 19 | 15 | 29 | 20 | 8 |
| | | General Improvement | 232 | 1 | | 1 | 3 | 2 | 6 | 1 | 5 | 5 | 11 | 7 | 8 | 12 | 14 | 7 | 12 | 13 | 14 | 15 | 11 | 7 | 14 | 19 | 11 | 16 | 14 | 3 |
| | Assessment and Ass. Models | CMMI | 170 | | 1 | | 1 | 1 | 1 | | 9 | 10 | 6 | 8 | 6 | 6 | 11 | 8 | 9 | 6 | 7 | 6 | 9 | 14 | 14 | 6 | 8 | 12 | 9 | 2 |
| | | ISO/IEC 15504 | 94 | | | | | 1 | | 1 | 7 | 8 | 5 | 7 | 4 | 5 | 6 | 2 | 3 | 5 | 2 | 5 | 3 | 5 | 3 | 1 | 5 | 12 | 4 | |
| | | General Measurement | 196 | 1 | 1 | 1 | | 2 | 4 | 4 | 8 | 8 | 4 | 6 | 8 | 9 | 10 | 6 | 8 | 8 | 7 | 12 | 12 | 13 | 11 | 12 | 9 | 14 | 14 | 4 |
| | (Quasi-) Standards (and Techniques) | Six Sigma | 13 | | | | | | | | | | | 1 | | | | | 2 | 1 | 2 | 3 | 1 | 1 | | | | 1 | 1 | |
| | | Bootstrap | 17 | | | | | | 1 | | 4 | 3 | 3 | 5 | | | | | | | | | | 1 | | | | | | |
| | | CompetiSoft | 4 | | | | | | | | | | | | | | | | | | | | | 1 | | | | | 2 | 1 |
| | | Continuous Improvement | 14 | | | | 1 | | | 1 | 1 | | | | 1 | 1 | 1 | | | | 1 | 1 | 1 | 1 | | 2 | 2 | | | |
| | | PSP/TSP | 17 | | | | | | | | | 1 | 1 | 1 | 2 | | 2 | 1 | | 1 | 1 | 2 | 2 | | 2 | 1 | | | | |
| | | ISO/IEC 29110 | 6 | | | | | | | | | | | | | | | | | | | | | 1 | | | | 1 | 3 | 1 |
| | | ISO/IEC 12207 | 7 | | | | | | | | | | | 1 | | | | | | | | | | 1 | | | | 2 | 3 | |
| Dimension: Study Type and Method | | Survey/Interview | 95 | 1 | | 1 | | | | | 5 | 2 | 1 | 4 | 4 | 4 | 8 | 4 | | 8 | 4 | 5 | 5 | 4 | 6 | 6 | 5 | 9 | 7 | 2 |
| | | Single Case-Study | 174 | | | 1 | | 1 | 3 | | 8 | 9 | 8 | 7 | 8 | 7 | 7 | 5 | 6 | 6 | 8 | 11 | 7 | 10 | 14 | 9 | 7 | 17 | 12 | 3 |
| | | Multi-Case/Long. Study | 136 | | | | 1 | 1 | 1 | | 1 | | 3 | 7 | 9 | 6 | 14 | 2 | 4 | 5 | 10 | 5 | 10 | 12 | 8 | 11 | 5 | 10 | 7 | 4 |
| | | Replication Study | 3 | | | | | | | | | | | | | | | | | | | | | 1 | | | | | 2 | |
| | | SLR/SMS | 55 | | | | | | 1 | | 1 | | 3 | 1 | 2 | 1 | | 1 | 1 | 1 | | 1 | 2 | 1 | 3 | 7 | 5 | 6 | 13 | 5 |
| | | Grounded Theory | 18 | | | | | | | | | | | | | | 1 | | | | 1 | 1 | 1 | | 1 | 2 | 3 | 3 | 4 | 1 |
| Dimension: Context | Life Cycle Phase | Project Management | 92 | | | | | | 2 | | 2 | 5 | 4 | 8 | 5 | 1 | 3 | 5 | 5 | 5 | 6 | 5 | 5 | 4 | 6 | 7 | 4 | 4 | 5 | 1 |
| | | Quality Management | 71 | | | 1 | | 1 | | 1 | | 3 | 3 | 2 | 4 | 3 | 3 | 4 | 4 | 4 | 1 | 7 | 2 | 4 | 1 | 7 | 3 | 5 | 5 | 3 |
| | | Requirements Engineering | 41 | | | 1 | | | 1 | | 3 | 1 | 1 | 2 | | | 2 | 3 | 3 | 5 | 5 | 2 | 2 | 3 | | | 3 | 3 | | 1 |
| | | Architecture | 17 | | | | | | | | | 1 | 1 | 1 | | | 3 | | 2 | 1 | 2 | 1 | | 1 | | 3 | | | 1 | |
| | | Implementation | 8 | | | | | | | | | | | 2 | | | 1 | | | | 1 | 2 | | 1 | | 1 | | | | |
| | | Test | 36 | | | | | | 1 | | 1 | | 2 | 1 | | 1 | 1 | | 2 | 2 | 2 | 1 | 1 | 2 | 4 | 3 | 2 | 6 | 4 | |
| | Company Size and Scale | VSE/SME | 116 | | | | | | | 1 | 1 | 2 | 3 | 5 | 2 | 7 | 8 | 2 | 7 | 4 | 4 | 6 | 5 | 8 | 15 | 7 | 6 | 7 | 12 | 4 |
| | | Other Company Size | 75 | 1 | 1 | 1 | | 1 | 4 | | 3 | 1 | 3 | 3 | 6 | 4 | 5 | 5 | 6 | 3 | 1 | 4 | 4 | 2 | 3 | 4 | 1 | 5 | 4 | |
| | | GSE | 37 | | | | | | 1 | | 2 | | | 2 | 2 | | 4 | 1 | | 4 | 1 | 2 | 3 | 4 | 5 | 1 | | 3 | 2 | |
| | Application Domain | Embedded Systems | 29 | | | | | | 1 | 1 | 1 | 1 | 2 | 2 | 3 | 1 | 1 | | 1 | 1 | | 1 | | 2 | 2 | 1 | | 3 | 5 | |
| | | Telecommunications | 23 | | | | | | 1 | | 1 | | 1 | | 4 | 3 | 2 | 2 | 2 | 1 | | 1 | 2 | 1 | 1 | 1 | | | | |
| | | Medical Devices | 10 | | | | | | | | | | | | | | | | | | | 1 | | | | | 2 | 5 | 2 | |
| | | Automotive | 14 | | | | | | | | | | | | | | | | 1 | 1 | | 1 | 3 | 2 | | 1 | 2 | 3 | | |
| | | Mission-critical Defense | 8 | | | | | | | | | 1 | 1 | 2 | | | 1 | 1 | | | | 1 | 1 | | | | | | | |
| | | Business IS | 9 | | | | | | | | | | | 1 | | | 1 | | | | 1 | | 1 | 1 | 1 | | | 2 | | 1 |
| | | Web/Mobile/Cloud | 11 | | | | | | | | | | | 1 | | 1 | 1 | | | 1 | | | 1 | | | 1 | 2 | | 2 | |
| | | Skills and Education | 17 | | | | | | | | 1 | | | | | | | | | 2 | | 1 | 1 | 4 | | 2 | 2 | | 1 | 3 |

**Figure 7** **Overview of the different metadata attributes addressed over time.** The darker the color, the more papers in a year have this attribute assigned, whereas a paper can have multiple attributes assigned.

by ISO/IEC 15504, which is assigned to 94 papers. Beyond the common standards, 196 papers are devoted to measurement in general. A more detailed discussion on the standard approaches CMMI and ISO/IEC 15504 can be found in 'New and customized SPI models'.

Regarding the *(quasi-)standards (and techniques)*, the overall result set indicates these aspects considered of low relevance for the community. Most frequently mentioned are Six Sigma, Continuous Improvement, and PSP/TSP (each with less than 20 mentions). Not yet clear is the relevance of standards like ISO/IEC 29110—we see some mentions, but there is some movement and continuous development of such standards. Therefore, a trend analysis is yet not meaningfully to conduct.

In the *publication objective* category, we analyzed the major research directive of a publication. Figure 7 shows four attributes in the spotlight: A considerable share of the papers (295 out of 769) deals with custom or new models, and the data shows the number of custom/new models continuously increasing. This trend, which was already found in the initial study, is discussed (together with the use of standard approaches) in 'New and customized SPI models'. Furthermore, 232 papers cover general improvement as a trend. Additionally, the result set contains 126 papers addressing SPI success factors with an increasing interest over the years. In 'SPI success factors,' we provide more details on this topic. Finally, with 73 mentions, agile and lean development constitutes the fourth trend with increasing number of publications. We provide details in 'SPI and agility'.

**Dimension: study type and method.** Within the six different attributes defined for this dimension, Fig. 7 shows single and multiple/longitudinal case studies the major instruments, followed by survey research and interview studies. However, as these instruments are often combined, i.e., in many case studies, data collection is carried out using interviews. Although the result shows so-called mixed method approaches applied to SPI research, still, single case studies (quite often carried out with students in lab environments) account for the majority of the selected research methods. Nevertheless, in recent years, an increasing number of secondary studies (i.e., systematic reviews and mapping studies) could be found. This indicates the community starting to systematize and categorize SPI knowledge. The result set clearly shows the research field lacking replication studies.

**Dimension: context.** Within the dimension *Context*, we defined the three categories *Life Cycle Phase*, *Company Size and Scale*, and *Application Domain*, which provide the following insights:

Regarding the *life cycle phases*, project management (92 mentions) and quality management (71 mentions) are in the spotlight (continuously covered and without specific peaks). They are followed by requirements engineering (41 mentions) and testing (36 mentions), whereas testing as topic is often combined with (general) quality management. Architecture and design as well as implementation received few mentions (less than 20 each).

The *companies sizes and scales* addressed in the papers show a trend towards very small entities (VSE) and small-to-medium-sized enterprises (SME). In the result set, 116 papers deal with companies of this sort, while 75 papers address companies of other scales, i.e., large companies and global players. In 'SPI for SMEs', we investigate this attribute group in more detail. Furthermore, global distribution of software development is addressed

by 37 papers, whereas this is a cross-cutting concern that is addressed by companies of all sorts. Considering the different *application domains*, the largest share of papers deals with embedded systems in general (29 mentions) or specific embedded domains such as medical devices, automotive software, or mission-critical and defense systems (less mentioned specific embedded domains are classified under general). The application domain of telecommunication systems is mentioned 23 times. We also consider the 17 papers addressing skills and education, e.g., by describing industrial training programs or university courses, as application domain.

**Summary.** Figure 7 presents an overview of the metadata attributes assigned to the 769 papers from the result set. The figure shows the major trends that we already observed in our initial study (*Kuhrmann et al., 2015*): SPI-related research has a strong focus on custom/new models and success factors, standard assessment/maturity models like CMMI or ISO/IEC 15504 are well-researched, and SPI in the context of VSEs/SMEs and agile and lean software development as part of SPI have to be considered major trends. The set of metadata attributes defined for this study provides further insights: for instance, major fields of interest in SPI research are project management and quality management (often in combination with testing), and SPI is relevant to all application domains and to all company sizes (which confirms *Horvat, Rozman & Györkös, 2000*). However, we also have to mention that due to the nature of this study, we were so far not able to assign attributes for all dimensions to all papers. Only 232 papers (30%) were assigned to attributes covering all three dimensions, 389 papers (51%) cover two dimensions, and 148 (19%) have attributes in only one dimension. Therefore, the presented overview does not yet provide a complete picture, and we discuss this threat to validity in 'Threats to validity'.

## RQ 3: trends in SPI-related research

Our initial study *Kuhrmann et al. (2015)*, inter alia, had the purpose to reveal trends in SPI-related research to identify those fields that have reached a certain saturation and those that either require more attention or reflect a particular problem-driven need. The initial results pointed to trends or streams worth further inspection: (new) SPI models, SPI success factors, SPI in small-to-medium-sized enterprises (SME), and agility as SPI. In subsequent sections, we primarily focus on these trends/streams, before discussing further observations.

### New and customized SPI models

In the field of SPI, existing (standard) models are customized or completely new models are proposed. This trend can be observed now for years, as Fig. 8 illustrates. Starting from the very beginning on, new or customized models are proposed every year. In total, the result lists 295 out of 769 papers (approx. 38%) with this purpose.

As shown in Fig. 2, in the present study, we collected metadata regarding different (quasi-) standard and well-disseminated approaches. In the following, we provide a detailed analysis on the share of customized and new models, and we analyze how these approaches are integrated with each other and what their scientific maturity is. Figure 9 shows a systematic map that illustrates two aspects: in lower part the research maturity and the contribution of papers addressing standard maturity models is shown. In total, 225 out

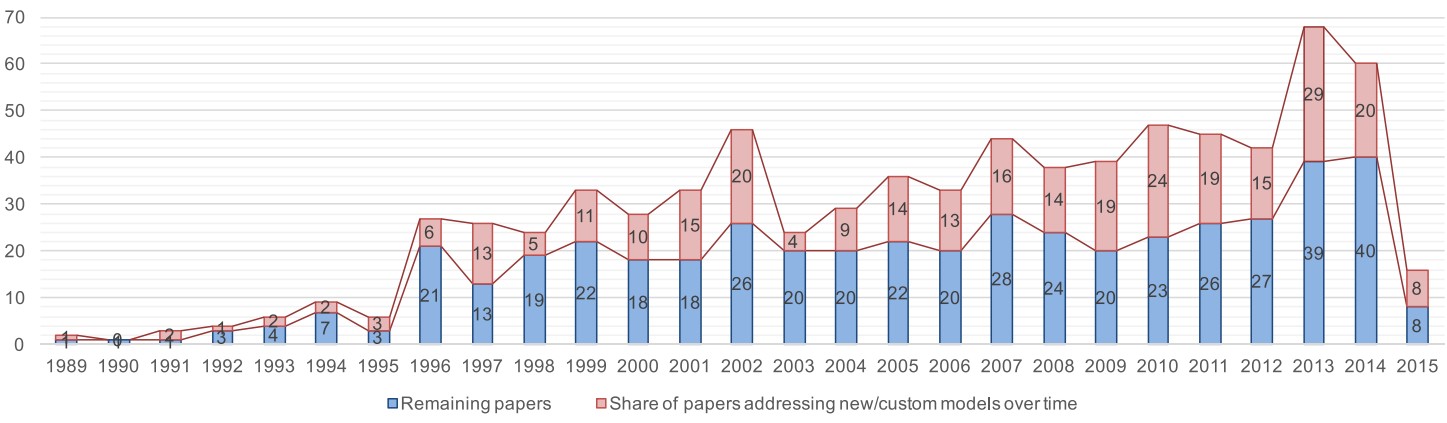

**Figure 8** Trend chart of the share of papers that present customized and/or new SPI models.

of 769 papers address CMMI, ISO/IEC 15504 or both. The classification according to the research- and contribution type facet shows that for standards and standard-related SPI research many lessons learned are reported and that some evaluation research is available.

From those 225 papers addressing standard approaches, 74 deal with developing customized SPI models, which are grounded in these standards. Whether a custom/new SPI model is based on one of the standards is visualized in the upper part of Fig. 9. From the 295 papers proposing custom/new SPI models, 74 are based on the standard models, i.e., 221 papers do not ground their contribution in standards and use other practices. In particular, four papers mentioned to reuse/extend Six Sigma, eight reused/extended the Continuos Improvement principle, three papers refer to PSP/TSP, and one paper refers to COMPETISOFT. Moreover, Fig. 9 shows that the result set contains 187 *solution proposals*, but only 76 papers that are categorized as *evaluation research* or *experience paper*. Among the 295 papers, 54 (18.3%) explicitly mention to cover SPI for SMEs (see also 'SPI for SMEs') with a focus on improving the project management (four papers) and general quality management processes (three papers). The processes associated with the different *life cycle phases* (Fig. 2) are represented as follows: 36 (12.2%) papers aim at improving the general quality management, 35 (11.9%) address project management, and 19 (6.4%) aim to improve the test process. That is, the focus of the custom/new SPI models is on quality management and testing (18.6% in total).

**Summary.** The trend observed in our initial study could be confirmed: 295 out of 769 papers propose custom or new SPI approaches, which makes in average 11 new SPI models published per year. Only 74 out of these 295 papers ground their contribution in a standard approach, whereas the majority (approx. 75%) of the solution proposals does not explicitly rely on standardized approaches. Furthermore, the result set shows that the majority of the papers proposing entire SPI methods or frameworks of which few are evaluated (the majority are *solution proposals*). Moreover, the result set shows few models or theories on SPI among the proposed solutions.

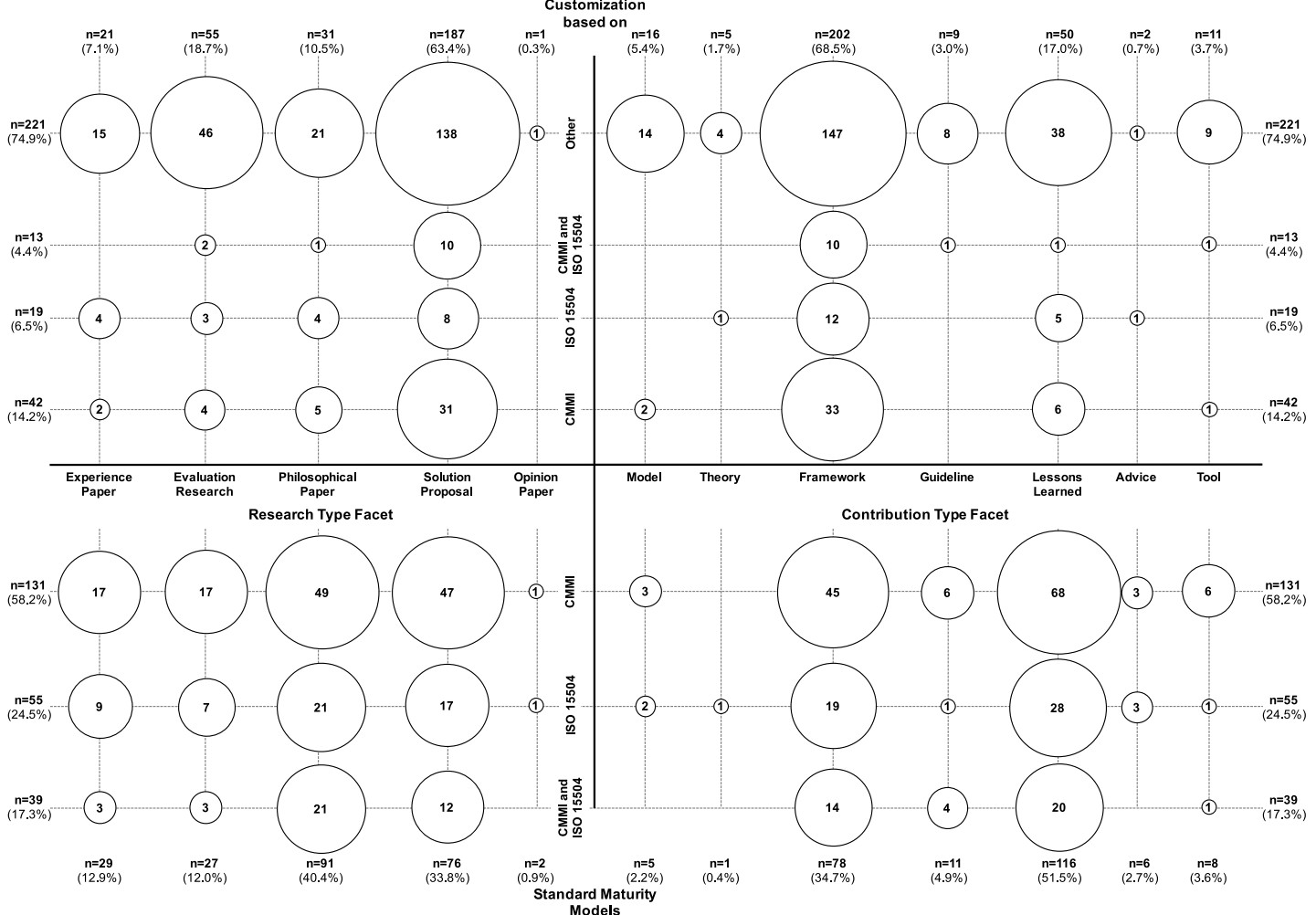

**Figure 9** Overview of the classification of publications addressing the standard approaches CMMI and ISO/IEC 15504 ($n = 225$), and their relation to custom/new models ($n = 295$).

### SPI success factors

Figure 10 visualizes the second trend observed: the quest for SPI success factors. In the result set, 126 out of 769 papers (approx. 16.4%) are devoted to success factors. The figure shows this quest starting in the mid 1990s, and an increasing interest starting around 2007. In the following, we provide an overview how success factors are collected, studied, applied, and evaluated.

The first questions of interest address the origin and maturity of the success factors, i.e., their general reliability. For this, we analyzed the research- and contribution type facets of the papers containing the success factors. Figure 11 provides this categorization and shows that 72 of the 126 papers (57.1%) are classified as *philosophical papers*, i.e., papers that are either a secondary study or that provide a discussion-based research approach. However, 33 papers (26.2%) derive their success factors either from evaluation research or experience reports. Furthermore, for 73 papers (57.9%), success factors are contributed

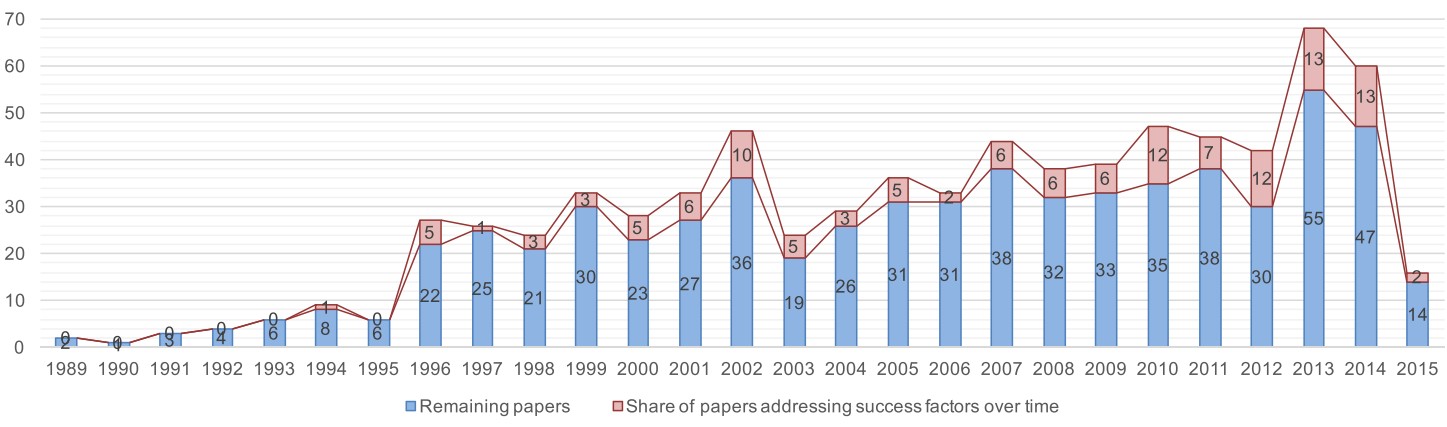

**Figure 10** Trend chart of the share of papers that investigate success factors in SPI.

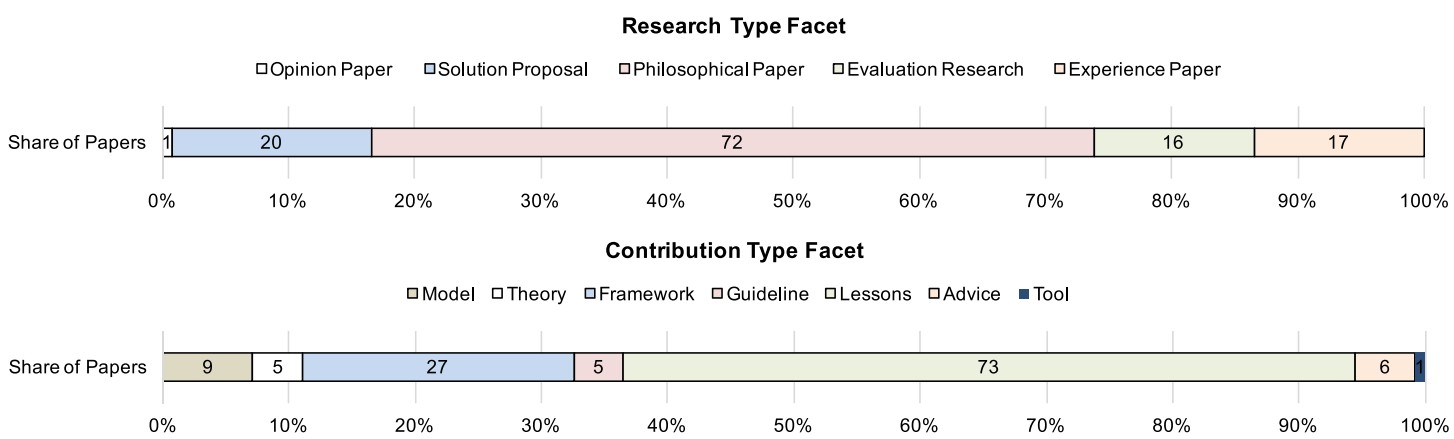

**Figure 11** Summary of papers addressing success factors in SPI categorized according the research- and contribution type facets.

as *lessons learned*; 27 papers (21.4%) structure and integrate success factors in *frameworks*, and 14 papers (11.1%) use success factors to develop a *model* or a *theory*.

Figure 11 suggests success factors mainly crafted from secondary studies and discussion. In order to provide more insight, we used the *Study Type and Method* dimension to study the research approaches chosen for the collection of success factors.

Figure 12 provides the summary of the chosen research methods. The figure shows survey/interview and case study research being the preferred methods. Only 18 out of 126 papers rely on secondary studies (systematic reviews and mapping studies), and only four papers use a multi-method research approach (either survey with case study research, or a secondary study combined with survey research and grounded theory). For 5 papers, an explicitly mentioned research approach could not be found in the abstract-based analysis. Figure 12 also shows that only 27 papers (21 multi-case/longitudinal study, 2 replication study, and 4 multi-method) go beyond "one-time research", i.e., these papers study success factors over time, from different angles, and/or apply them and learn from the application.

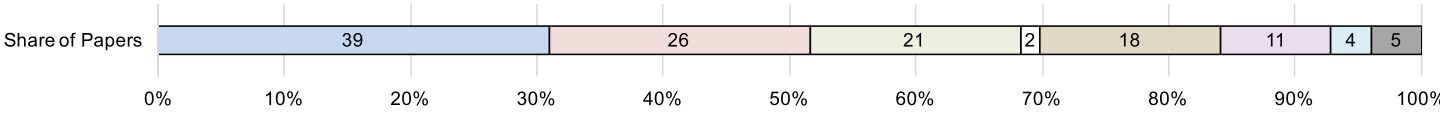

**Figure 12  Summary of the research methods applied to study SPI success factors.**

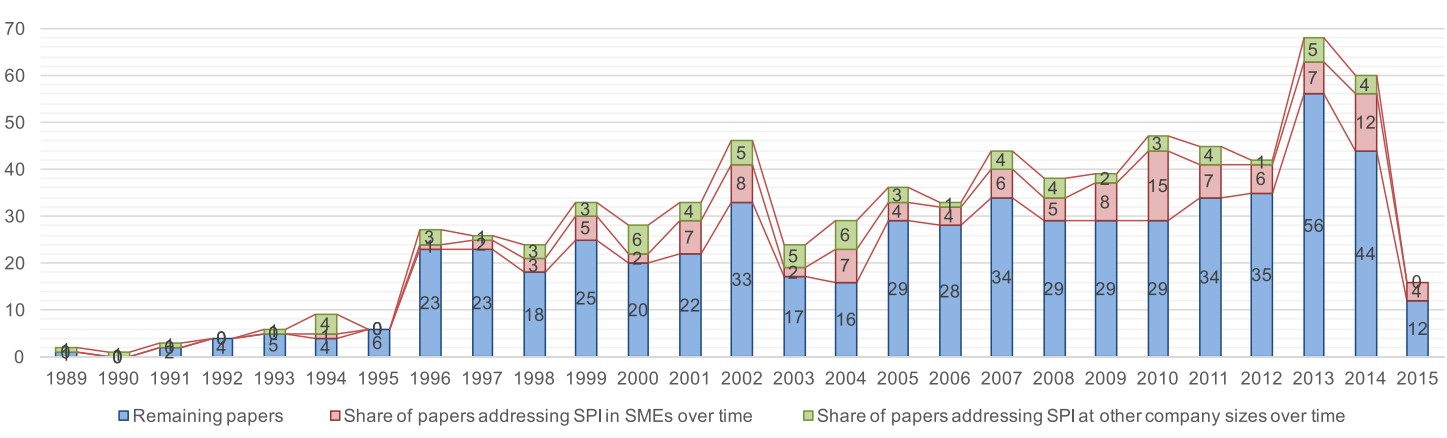

**Figure 13  Trend chart of the share of papers SPI in the context of SMEs.**

**Summary.** The second trend observed in our initial study could be confirmed: 126 out of 769 are devoted to the collection and study of success factors. The majority of the papers is classified as *philosophical papers*, i.e., these papers report secondary studies or discussion-based studies, and most of the papers present success factors as *lessons learned*. However, the data also indicates success factors being crafted from limited research in terms of long-term observation or evaluation from different angles. Only 27 papers mention a respective research approach. Furthermore, 18 out of 126 papers are categorized as secondary studies, i.e., there is an observable trend to foster information collection and aggregation.

### SPI for SMEs

The third trend observed in the initial study was an increasing interest in SPI for small-to-medium-sized enterprises (SME). Figure 13 provides an overview of the share of papers explicitly addressing SPI in SMEs (and other company sizes if mentioned in title, keywords, or abstracts).

The figure shows a first "peak" from 1996-2002 (matching the "dot-com" phase), and then a growing interest starting again in 2007 continuing till now. In total 186 out of 769 papers explicitly mentioned the company size in the context attributes of which 116 papers (15.1%) mention SMEs (or VSEs), and another 75 papers (9.8%) mention other company sizes; one paper addresses companies regardless of their size. Cross-cutting the company size, the metadata also contains an attribute for *Global Software Engineering* (GSE), i.e., if SPI takes place in a global setting. A total of 37 papers address GSE-related questions. In

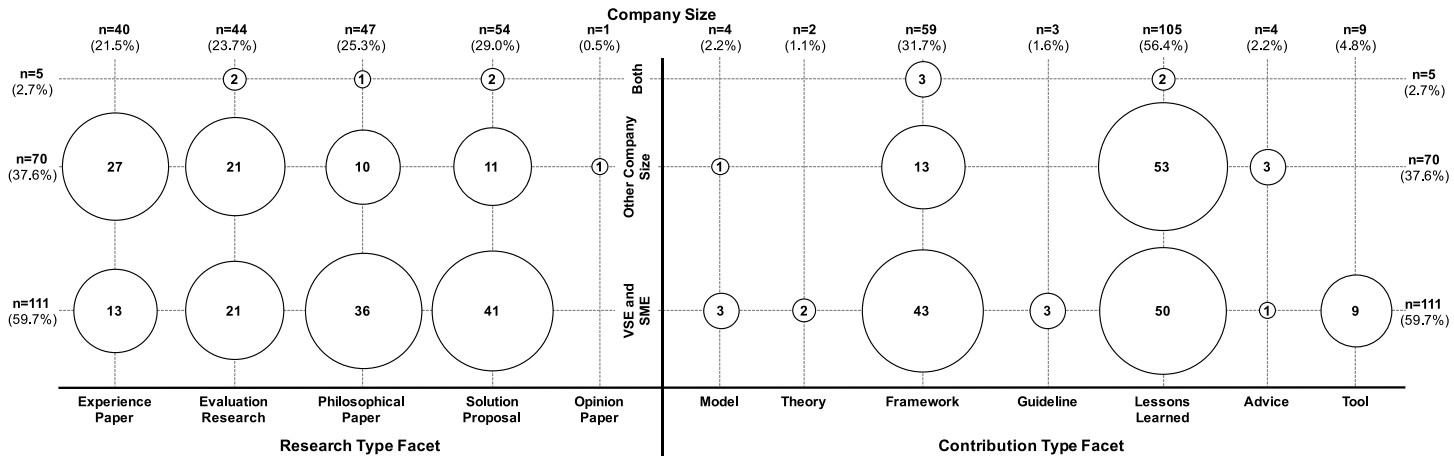

**Figure 14  Overview of the classification of publications addressing SPI in small and very small companies, and SPI in other company sizes (*n* = 186).**

the following, we provide some insights regarding the topics SPI for SMEs addresses and we also provide an overview of the respective application domains and covered life cycle phases.

Figure 14 provides a systematic map of the papers that explicitly mention the company context. The figure shows the classification according to the research- and contribution type facet. Regarding the research type facet, Fig. 14 shows a fairly balanced picture, i.e., we find *solution proposals*, *philosophical papers*, *evaluation research*, and *experience papers*. Regarding the contribution type facet, papers mostly provide frameworks and lessons learned. However, for VSEs and SMEs, three papers develop models on SPI for SMEs, two papers develop theories on SPI for SMEs, and nine papers also address tools in the context of SPI for SMEs.

The get more insights, we filtered the metadata for the company size. The results are illustrated in Tables 6–8. Table 6 shows that most of the VSE/SME-related papers emerge from the domain of web, mobile, and Cloud-based software development. Companies categorized as "other," i.e., large companies and global players, mostly contribute to the body of knowledge from embedded systems and telecommunication. Regarding the respective publication objectives, Table 7, again, shows the trend to contribute custom/new SPI models—especially for the VSE/SME context (cf. 'New and customized SPI models'), and to collect success factors (cf. 'SPI success factors'). Table 7 also shows the interest into agile and lean approaches in the context of SPI. As already mentioned in 'New and customized SPI models', a certain trend shows a particular focus on improving project- and quality management. Table 8 reflects this trend also for the company-size context, whereas large companies and global players seemingly address a broader spectrum of life cycle phases.

**Summary.** Among the 769 papers from the result set, 186 explicitly mention the company size as context attribute. In total, 116 papers explicitly mention small and very small companies as research context. Almost half of the papers (54 papers) address custom/new SPI models, which confirms the previously observed trend. In the present result set, we

**Table 6** Overview of SPI application domains.

| Application domain | V/SME | Other |
|---|---|---|
| Embedded system | 1 | 9 |
| Telecommunication | 0 | 16 |
| Medical devices | 0 | 0 |
| Automotive | 2 | 1 |
| Mission-critical defense | 1 | 4 |
| Business IS | 1 | 4 |
| Web/Mobile/Cloud | 8 | 1 |
| Skills and education | 1 | 1 |

**Table 7** Overview of publication objectives.

| Publication objective | V/SME | Other |
|---|---|---|
| Agile/Lean | 9 | 7 |
| Process simulation | 0 | 1 |
| Process Line/Patterns | 1 | 2 |
| Product Line/Management | 1 | 1 |
| Success factors | 21 | 8 |
| Custom model | 54 | 23 |
| General improvement | 29 | 28 |

**Table 8** Overview of addressed life cycle phases.

| Life cycle phase | V/SME | Other |
|---|---|---|
| Project management | 13 | 10 |
| Quality management | 6 | 7 |
| Requirements engineering | 1 | 6 |
| Architecture | 3 | 4 |
| Implementation | 2 | 2 |
| Test | 1 | 4 |

find a growing interest in SPI for SME, which is also supported by the recently published standard ISO/IEC 29110 that explicitly addresses SPI for small and very small companies (six papers already refer to this new standard).

### SPI and agility

Finally, Fig. 15 visualizes the fourth trend found in the initial study: although perceived as contradiction, in recent years, combining agility and SPI received some attention, such as agile maturity models. In total, the result set contains 73 papers (9.5%) that address agility in the context of SPI, and the Fig. 15 shows first contributions on this topic just around the Agile Manifesto's publication. However, the "real" interest started around 2008, similar to *Salo & Abrahamsson (2007)*, when the number of studies dealing with agility and SPI started to increase.

Figure 16 shows the big picture by visualizing the research- and contribution type facets of the papers on agility and SPI. The figure shows a balanced research, i.e., the

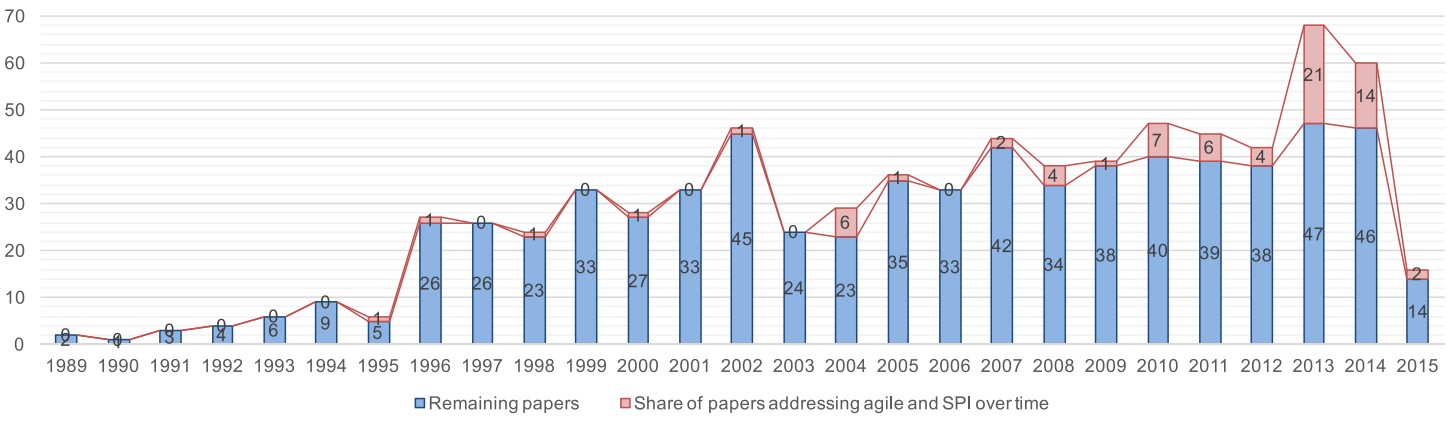

**Figure 15** Trend chart of the share of papers that investigate the application of agility in SPI.

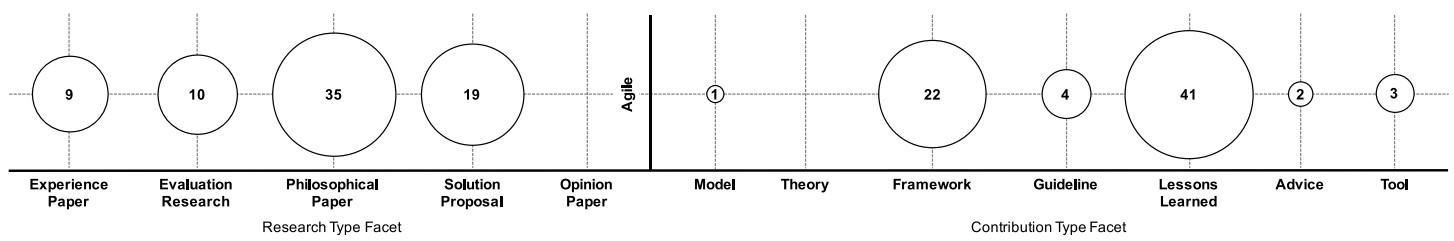

**Figure 16** Overview of the classification of publications addressing agility and SPI ($n = 73$).

result set contains *solution proposals* as well as *evaluation research* and *experience reports*, and *philosophical* papers discussing agility and SPI (only two of the *philosophical* papers are secondary studies). The majority of the 73 papers contributes *lessons learned* (from applying agile in SPI or related activities) and *frameworks*.

Analyzing the 73 papers for the collected metadata, 20 papers discuss agility in the context of the standard SPI models, i.e., CMMI and ISO/IEC 15504. Furthermore, 22 papers propose custom SPI models of which six papers ground their proposal in CMMI and three papers in ISO/IEC 15504. 16 papers discuss success factors associated with agility and SPI, whereas only one paper develops a model on success factors while of the remaining papers 12 report lessons learned only. Regarding the company size, nine papers explicitly mention VSEs and SMEs as research context and seven papers address other company sizes (mostly in the embedded systems and telecommunication application domain). Furthermore, five papers discuss agility in a Global Software Engineering context (three of them in the context large companies). Finally, regarding the covered life cycle phases, six papers aim to improve the project management and nine papers address quality management and software test.

**Summary.** Among the 769 papers from the result set, 73 deal with agility and SPI. These papers address a variety of topics showing agility considered relevant for many aspects of software and system development thus becoming interesting for SPI, too. The majority of the classified papers deals with agility as concept to improve processes. However, the result set also contains papers adapting agility for SPI as such, like agile maturity models

(e.g., *Schweigert et al., 2013*) or concepts to justify agility and standard SPI models. The result set also shows that agility is not for V/SMEs only, but also large companies and even global players have a growing interest into agility.

## Discussion

In this section, we discuss the findings obtained so far. Beyond the discussion of the trends already identified in our initial study, we also broaden our perspective and discuss further trends that can be found in the updated result set.

**Further insights in SPI research.** Beyond the aforementioned major trends, the updated study (including the updated data analysis procedures) reveals more insights but few further trends. At first, the study confirms the statement by *Horvat, Rozman & Györkös (2000)* that SPI is important for all companies regardless of their size, and, we can add, also regardless of their application domain. Rationale for this growing interest can be found in new technologies and markets (see also attribute GSE in Fig. 7), and in the evolution of software development methods. For instance, several studies like the "State of Agile Survey" (*VersionOne, 2006–2014*) show a growing interest in agile and lean approaches and, at the same time, *Vijayasarathy & Butler (2015)* and *Theocharis et al. (2015)* study how this trend is manifested in the companies' process use. Especially *Theocharis et al. (2015)* mention *hybrid software processes* (or the "Water-Scrum-Fall" as named by *West, 2011*) as standard approach. Yet, so far, little is known about the (systematic) development of such hybrid processes. This can be considered one reason for the growing interest in SPI: companies want/have to adopt agile/lean approaches (e.g., *Diebold et al., 2015*), but they also have to comply with external norms and standards (e.g., in the domain of safety-critical systems), which we consider a main driver behind SPI initiatives. Another perspective is given by VSEs and SMEs that also have a growing interest in SPI. However, for companies of this size, standard approaches, such as CMMI or ISO/IEC 15504 are often inappropriate (see for instance *Staples et al., 2007*). At this scale, agile/lean is important as well as context-specific SPI approaches, which can be considered an explanation for the significant number of custom/new SPI models ('New and customized SPI models') such as LAPPI (*Raninen et al., 2012*) or tailored standards, such as ISO/IEC 29110 (*Laporte & O'Connor, 2014*).

Another finding of the study is a strong focus on project management and quality management (often together with testing) in SPI. SPI is, usually, a management-driven endeavor. As argued in *Theocharis et al. (2015)*, managers want to have their "safe" and measurable environment, while developers prefer slim and agile development approaches (see also *Murphy et al. (2013)*; *Tripp & Armstrong (2014)*). This line of argumentation provides rationale for two observations from this study: first, there is a continuous effort in studying measurement in general and, second, the growing interest in agile/lean approaches. Both together lead to a number of the aforementioned hybrid software processes and also to context-specific SPI approaches that—all together—provide an explanation for the strong focus on project- and (general) quality management. Regarding the remaining life cycle phases, requirements engineering and software test are the most frequently researched topics in SPI. However, the high number of testing-related papers (compared to the implementation-related papers) motivates the question for why this rather "late"

phase is more emphasized, especially in times of agile/lean software development. Is testing addressing implementation as well? Is testing subject to improvement because of the effort spent on this activity? However, these are questions that cannot be answered in the current stage of the study thus remain subject to future work (see also 'Conclusion & Future Work').

**What is the state of SPI after all?** Our data shows a diverse picture and, furthermore, shows SPI a frequently researched topic (Fig. 3). Moreover, research on SPI addresses a variety of aspects with certain focus points: The majority of the investigated publications focuses on proposing custom/new frameworks and on reporting lessons learned. Furthermore, our results show a significant imbalance between proposing new solutions and evaluating their feasibility—especially in the long run. The majority of evaluation research is conducted in the context of standardized SPI- and maturity models (Fig. 9). For newly proposed models, we often find—if at all—only single-case validation (in industry or university-hosted labs); only few, e.g., *Raninen et al. (2012)* provide a comprehensive evaluation. Another finding is the lack of theorizing approaches, which are often performed for specific domains (e.g., SMEs) or grounded in secondary studies only. In summary, although SPI is around for decades, we still miss a sound theory about SPI. We have a number of standardized and specific SPI models and frameworks. However, we still lack evidence.

One reason could be that SPI always involves change in behavior of individual persons and changes in the culture of an organization. Due to the varying contexts, SPI cannot be too descriptive. Therefore, frameworks and tools are proposed for adaptation to the respective context. This would also provide an explanation for the effort spent to study SPI success factors ('SPI success factors'), which can be considered an early step towards crafting a more general and context-agnostic theory on SPI. Yet, the constant change or evolution of the context could be considered a continuous stimulus to provide new frameworks that only have a short life cycle and are quickly replaced by other frameworks that aim to "better" solve a particular issue. This assumption is supported by the missing long-term and replication studies (the result set only contains 2 explicitly mentioned replication studies). Yet, this constant change could also put all attempts to standardize SPI at stake. As for instance *Vijayasarathy & Butler (2015)* and *Theocharis et al. (2015)* have shown, companies utilize highly customized and specific processes, and the aforementioned diversity could end up in a situation in which every organization implements its own "home-grown" SPI approach, leaving only non-binding initiatives, such as the SPI Manifest (*Pries-Heje & Johansen, 2010*) as least common denominator.

Furthermore, missing is a critical discussion and comparison of available approaches, and their use and feasibility in practice. Although we found 55 secondary studies, these studies lay their focus on investigating success factors rather than providing structure and trying to generalize available knowledge, as for instance done by *Unterkalmsteiner et al. (2012)*. However, in our study, we found more than 200 papers addressing standard SPI approaches, 295 papers presenting/discussing custom/new models, and we also found 126 papers explicitly devoted to SPI success factors. All together, these papers provide a rich ground to conduct research on the evolution of SPI models, which would help studying the actual essence of SPI models, factors that positively/negatively influence the success of

SPI programs. In a nutshell, our results show that SPI is a still emerging field characterized by solution proposals and experiences awaiting more effort to systematization.

## Threats to validity

In this section, we evaluate our findings and critically review our study regarding its threats to validity. As a literature study, this study suffers from potential incompleteness of the search results and a general publication bias, i.e., positive results are more likely published than failed attempts. For instance, the result set does not contain studies that explicitly report on failure and draw their conclusions from respective lessons learned, and we thus cannot analyze proposals to answer the question for: What works and what does *not*? That is, our study encounters the risk to draw an incomplete and potentially too positive picture.

**Internal validity.** Beyond the aforementioned more general threat, the *internal validity* of the study could be biased by personal ratings of the participating researchers. To address this risk, we continued our study *Kuhrmann et al. (2015)*, which follows a proven procedure *Kuhrmann, Fernández & Tiessler (2014)* that utilizes different supporting tools and researcher triangulation to support dataset cleaning, study selection, and classification.

Furthermore, due to the inappropriateness of the *focus type facet* as classification schema in this stage of the study (as already discussed in *Kuhrmann et al., 2015*), we addressed this threat to validity by relying on a new, more flexible set of metadata ('Analysis and classification'). This new instrument addresses the previously found issues, namely (general) disagreement on the categorization, and lacking precision and demand for multiple assignments respectively. However, although the issues with the focus type facet were solved, the metadata schema introduces potentially new threats. For instance, due to the nature of the study, we cannot ensure to have a full set of metadata for every paper (as already mentioned in 'RQ 2: result set contribution', only 30% of the papers have attributes from all three metadata dimensions assigned and, still, we cannot ensure to have captured all metadata). Furthermore, the metadata collected so far needs to be considered *initial*, as there are potentially more attributes of interest. That is, since we rely on the mapping study instrument in the first place, some metadata might yet not be captured, as this would require a more in-depth analysis, e.g., using the systematic review instrument. Furthermore, as we introduced 40 metadata attributes, the risk of misclassification increases, e.g., due to misunderstandings regarding the criteria to be applied or due to confusing/misleading use of terminology in respective papers.

**External validity.** The *external validity* is threatened by missing knowledge about the generalizability of the results. However, as we focused on a broadband analysis accepting a large number of publications, we assume to have created a generalizable result set. Furthermore, due to an extra quality assurance and trend analysis of the two result sets (initial study and study update) and the integrated result set, in 'Result overview', we could observe a manifesting trend (see Figs. 3–5). Yet, this assumption needs to be confirmed by further independently conducted studies. Also, the external validity can be threatened by the modified data collection procedure (Appendix 'Data collection in the study update'), which includes a potential limitation of the update chunks to be added. However, the

aforementioned quality assurance and trend analysis procedures did not show a significant impact on the trends of the distribution of the papers in the result sets.

Nevertheless, to increase the external validity, further update and/or replication studies are required to confirm our findings. With the study at hand, we lay the foundation for such research by providing an actionable update procedure (Appendix 'Search and cleaning procedure') that can be implemented by further researchers. Furthermore, as already mentioned in the discussion on the internal validity, generalizability is also affected by potential white spots in the metadata attributes, which, however, requires further investigation. Such (independently conducted) investigation will (i) contribute to the internal validity by increasing dataset completeness, but (ii) will also improve the external validity by incrementally improving the quality of the dataset used to draw general conclusions.

## CONCLUSION & FUTURE WORK

In this article, we presented a substantially updated systematic mapping study on the general state of the art in Software Process Improvement (SPI). The present work continues our long-term study of which we published initial results in *Kuhrmann et al. (2015)*, and (i) evolves the dataset and the precision of the data analysis and (ii) introduces an improved data collection instrument to serve further studies of the field. To analyze the data obtained from automatic searches, we rely on the research type facet by *Wieringa et al. (2005)* and the contribution type facet by *Shaw (2003)* as standard classification schemas. Furthermore, to get deeper insights, we defined 40 metadata attributes. In total, our study results in 769 papers that allow for a long-term analysis of the development of SPI, and that allow for determining research hot-spots and (general) trends.

In particular and based on *Kuhrmann et al. (2015)*, our study investigates previously observed trends: a constant publication rate of custom/new SPI models, a huge interest into studying SPI success factors, and an increasing interest in studying SPI in the context of (very) small enterprises and in adopting agile principles and practices to SPI. Among other things, 295 papers (38%) of the papers propose/discuss custom or new SPI approaches (ranging from fully-fledged models to specific fine-grained methods). From these 295 papers, 74 ground their contribution in standard models, such as CMMI or ISO/IEC 15504, whereas the majority of the papers is based other practices or none of the available approaches. The majority of the custom/new models covers self-contained SPI approaches, which are, however, scarcely evaluated in a broader context (the most frequently used instrument to conduct SPI research is the single-case study). Moreover, the publication pool is focused on solution proposals, yet lacking theories or models of SPI. Regarding the second trend, 126 papers (16.4%) were identified contributing SPI success factors. The investigation of how the success factors were distilled showed an increasing trend towards secondary studies. That is, although most of the contributing papers report on rather short-term studies or studies carried out in a university lab (only 27 papers mention a mixed-method or long-term research approach to investigate and evaluate success factors), there is an observable trend to foster information collection and structuring. The third trend

is the increasing interest into SPI in the context of VSEs and SMEs. In the result set, 116 papers (15.1%) explicitly address companies of this size of which about the half (54 papers) addresses custom/new SPI approaches tailored to this particular context. Yet, the result set also shows new standards that address this context (e.g., the ISO/IEC 29110) represented in the study. The last trend studied addresses agility and SPI. The result set mentions 73 papers (9.5%) mostly using agility as a concept to improve established processes, but the result set also lists agile maturity models or further concepts to justify agility and standard SPI models. The result set also shows that agility is not for VSEs/SMEs only, but also large companies and even global players, e.g., from the domain of telecommunications, show a growing interest into agility. Finally, going beyond the aforementioned general trends, inspecting the result set shows SPI mostly addressing project management and quality management (including measurement), and the result set shows the growing interest into agile/lean approaches.

**Impact.** Summarizing, our study provides a big picture illustrating the development of the field SPI over more than 25 years. Our results show a diverse picture, which is shaped by a constant publication rate of about 11 SPI solution proposals per annum, and a large share of papers reporting lessons learned. However, our study also shows an imbalance in the publication pool: there are many solution proposals but few are rigorously evaluated. Furthermore, although SPI as a field addresses a variety of topics, on the one hand, our study shows several research hotspots but, on the other hand, we could also identify "under-researched" topics, such as sound theories and models on SPI.

Therefore, our study has some impact on research as well as on practice. From the practitioner perspective, by using the categorized data, our study helps practitioners better characterize an actual/planned SPI endeavor and to find proper approaches and experiences straight forward and thus helps avoiding errors already made before or re-inventing the wheel. For researchers, our study provides rich ground to conduct further research, e.g., by highlighting the white spots that need further investigation or by naming those fields that already accumulated a certain amount of data thus enabling researchers to conduct replication research.

**Limitations.** Although being a long-term endeavor aggregating much knowledge, our study has some limitations. In particular, due to the overall goal of creating the big picture, our study suffers from the mapping study instrument applied. As a mapping study, our study suffers from missing details and, therefore (as discussed in the threats to validity), bears the risk of incomplete or even incorrect data classification. However, to overcome this major limitation, further (independently conducted) research is required to incrementally improve the data. Furthermore, the present study is conducted from the perspective of "pure" SPI. That is, (very) specific SPI approaches in specific domains might not be triggered by the study design. To overcome this limitation, again, further complementing research is required to improve the data quality.

**Future work.** Addressing the aforementioned limitations of the present study, future work comprises a collection of fine-grained studies for selected aspects. In particular, the study presented here serves as a *scoping study* to identify certain hotspots, trends, or streams

worth further investigation. Based on those hotspots, we form data subsets, which we analyze using the systematic review instrument (instead of the mapping study instrument) to conduct in-depth analyses. Currently, we called in further external researchers to strengthen the team and to carry out the following in-depth studies on SPI in the field of Global Software Engineering (GSE; *Kuhrmann et al., in press*), SPI in the context of software quality management and testing, agility and SPI, and SPI barriers and success factors. Conducting these studies helps rounding out the big picture and, moreover, to get more details and insights on specific topics of interest. Furthermore, by applying the systematic review instrument, we directly address the aforementioned limitation and incrementally improve the data quality. In further iterations of the main study, such improved data is going to be integrated with the main study thus aiding the general improvement of the data and analyses presented here. As the present study is also designed to serve as a continuous measurement of SPI's heartbeat, the next update of the mapping study (including all detailed data obtained by then) is planned for 2017.

## ACKNOWLEDGEMENTS

Conducting such a study is a long-term endeavor that so far involved many people. We thank Michaela Tießler, Ragna Steenweg, Daniel Méndez Fernández for their support in the early stages of this study, in particular in testing the instruments for paper selection and dataset cleaning. Furthermore, we owe special thanks to the students of the ''Hiwi-Pool'' of the Technische Universität München, who supported us during the initial data collection and dataset completion and cleaning processes, and we also owe special thanks to Claudia Konopka and Peter Nellemann, who substantially supported the initial data analysis and reporting. Furthermore, we thank the reviewers and participants of the *International Conference on Software and Systems Process* (ICSSP) 2015 for their valuable comments and the inspiring discussion on the initial study results.

## APPENDIX A. INITIAL STUDY POPULATION

In the initial study, based on the data collection procedures (described in Appendix 'Data collection in the initial study') and the study selection procedures (described in 'Research Design'), we obtained the result set described in Table 9. This dataset is the foundation for *Kuhrmann et al. (2015)*, and this result set also lays the foundation for the study update presented in this paper.

## APPENDIX B. DATA COLLECTION PROCEDURES

The presented study lays the foundation for a continuous study of the research field of *Software Process Improvement* (SPI). In order to support this long-term study, an efficient *study update procedure* is an imperative, which mainly affects the data collection procedures. Therefore, in this appendix, we give an integrated and detailed view on the data collection procedure as executed in the initial study, and we detail the update procedure used for compiling the report at hand.

**Table 9** Data collection and filtering results (tentative result sets during selection and final result set).

| Step | IEEE | ACM | Springer | Elsevier | Wiley | IET | Total |
|---|---|---|---|---|---|---|---|
| *Step 1: Search ('Query construction')* | | | | | | | |
| $S_1$ **and** $(C_1$ **or** $C_2)$ | 71 | 543 | 306 | 991 | 1,185 | 89 | 3,185 |
| $S_2$ **and** $(C_1$ **or** $C_2)$ | 68 | 539 | 306 | 989 | 1,133 | 89 | 3,124 |
| $S_3$ **and** $(C_1$ **or** $C_2)$ | 1,310 | 2,341 | 1,032 | 2,675 | 16,113 | 726 | 24,197 |
| $S_4$ **and** $(C_1$ **or** $C_2)$ | 130 | 925 | 438 | 945 | 2,480 | 479 | 5,397 |
| $S_5$ **and** $(C_1$ **or** $C_2)$ | 1,585 | 2,459 | 1,038 | 2,731 | 17,184 | 822 | 25,819 |
| $S_6$ **and** $(C_1$ **or** $C_2)$ | 535 | 1,746 | 762 | 1,863 | 9,182 | 484 | 14,572 |
| $S_7$ **and** $(C_1$ **or** $C_2)$ | 168 | 324 | 143 | 242 | 765 | 41 | 1,683 |
| $S_8$ **and** $C_2$ | 114 | 105 | 433 | 1,015 | 6,341 | 366 | 8,374 |
| Step 2: Removing Duplicates ('Analysis preparation') | | | | | | | |
| Duplicates per database | 1,486 | 566 | 4,388 | 7,161 | 1,328 | 1,714 | 16,643 |
| Duplicates across all databases | 916 | 551 | 1,059 | 2,043 | 370 | 376 | **5,315** |
| *Step 3: In-depth Filtering ('Filter queries')* | | | | | | | |
| Applying filters $F_1$ and $F_2$ | 578 | – | – | 710 | 221 | 53 | 1,562 |
| Unfiltered | – | 551 | 1,059 | – | – | – | 1,610 |
| Result set (search process) | 578 | 551 | 1,059 | 710 | 221 | 53 | **3,172** |
| *Step 4: Voting ('Analysis preparation')* | | | | | | | |
| **Final result set** | 283 | 65 | 114 | 103 | 67 | 3 | **635** |

## Data collection in the initial study

The initial study, inter alia, aimed at creating the baseline to study SPI. Therefore, the initial study was carried out with a considerable "manpower" that, however, is too costly for a continuous update. In this section, with the purpose of increasing transparency and reproducibility, we present the details of the initial data collection procedure (see also *Kuhrmann et al., 2015*), before presenting the implemented—and recommended— approach to conduct the study updates in Appendix 'Data collection in the study update'.

### Query construction

In a series of workshops, we defined the keywords that we are interested in and defined the general search strings in Table 10, which were then validated in several test runs before being used in an automated full-text search in several literature databases. The queries were built based on keyword lists given by the common terminology in the area of software processes and SPI.

**General queries.** The general search strings $S_1$–$S_8$ were defined according to the relevant topics in SPI, e.g., improvement, assessment, measurement, ISO/IEC 15504, CMMI, quality management, and so forth. Due to the expected large number of results, we decided to complement the general search strings with context selectors $C_1$ and $C_2$ to limit the search to the domain of interest. Finally, we concluded the search strings shown in Table 10.

**Filter queries.** Because of the full-text search, we expected a variety of publications including some overhead. Hence, we defined two filter queries $F_1$ and $F_2$ to be applied to the initial result set with the purpose of reducing the result set to the key publications. Query

**Table 10   Search strings used for the database search in the initial study *Kuhrmann et al. (2015)*.**

| | Search string | Addresses… |
|---|---|---|
| $S_1$ | (life-cycle or lifecycle or life cycle) and (management or administration or development or description or authoring or deployment) | process management: general life cycle |
| $S_2$ | (life-cycle or lifecycle or life cycle) and (design or modeling or modelling or analysis or training) | phases of the software process's life cycle |
| $S_3$ | modeling or modelling or model-based or approach or variant | process modeling |
| $S_4$ | optimization or optimisation or customization or customisation or tailoring | process customization and tailoring |
| $S_5$ | (measurement or evaluation or approach or variant or improvement) | general measurement and improvement |
| $S_6$ | reference model or quality management or evaluation or assessment or audit or CMMI or Capability Maturity Model Integration | reference models and quality management |
| $S_7$ | SCAMPI or Standard CMMI Appraisal Method for Process Improvement or SPICE or ISO/IEC 15504 or PSP or Personal Software Process or TSP or Team Software Process | reference models and assessment approaches |
| $S_8$ | (feasibility or experience) and (study or report) | reported knowledge and empirical research |
| $C_1$ | software process and (software development model or process model) | *context definition:* software processes |
| $C_2$ | SPI or software process improvement | *context definition:* SPI |
| $F_1$ | (SPI or software process improvement) and (approach or practice or management) | SPI approaches, practices, and SPI management |
| $F_2$ | (SPI or software process improvement) and report and (feasibility or experience) | evaluation research on SPI, e.g., studies, reports, etc. |

$F_1$ aims at finding all publications in the result set that explicitly present SPI approaches and practices, or that address the management of SPI. $F_2$ aims at finding all reports in the context of SPI in which feasibility is analyzed or experiences are reported. While the initial search was a full-text search, the filter queries were applied to the abstracts only. However, for technical reasons, ACM and Springer abstracts were partially not available in the initial result set and, thus, the filtering was done manually during the voting procedure (cf. Appendix 'Analysis preparation').

### Data sources and data format

The initial data collection was an automated full-text search in several literature databases. As main data sources, we relied on established literature databases, which we consider most appropriate for a search. In particular, we selected the following databases: *ACM Digital Library*, *SpringerLink*, *IEEE Digital Library* (Xplore), *Wiley*, *Elsevier* (Science Direct), and *IET Software*. If there was a paper listed in one of those databases, but was only referred, we counted it for the database that generated the item, regardless of the actual publication location.

### Analysis preparation

We performed an automated search that required us to filter and prepare the result set. The data analysis is prepared by harmonizing the data and performing a 2-staged voting process.

**Harmonization.** Due to the query construction, we found a vast amount of multiple occurrences in the result set, and we also found a number of publications that are not in software engineering or computer science. To make the selection of the contributions more efficient, we first cleaned the initial result set (cf. Table 9 for the results per phase). In the first step, we removed the duplicates, which we identified by title, year, and author list. In the second step, we applied the filter queries to sort out those publications not devoted to software processes and SPI. To double-check the result set, we used *word clouds* generated from abstracts and keyword lists to validate if the result set meets our requirements.[3] This procedure was performed individually per database and again on the integrated result set. Finally, we completed missing data to prepare the voting procedure.

**Voting the papers.** The final selection whether or not a paper was included in the result set was made using a multi-staged voting procedure. This procedure was also applied in the study update and, therefore, is described in detail in 'Analysis preparation'.

## Data collection in the study update

In this section, we present the details about the recommended data collection procedure to be implemented for study updates.

### Search queries

The major update in the search procedure is the search engine utilized for the search. Instead of repeating the search with individual databases (cf. Appendix 'Data sources and data format'), we switched to Scopus, as Scopus as meta-search engine covers most of the relevant software engineering venues (journals as well as conferences). This however changes the general search procedure, notably the search strings need to be updated accordingly. The adapted search strings are summarized in Table 11. Comparing the new search queries to the initial study's queries from Table 10, it becomes obvious that the context selectors and filter queries are now integrated with the search strings. We tested the new search queries several times on subsets of the initial study before executed them to carry out the actual data collection.

### Search and cleaning procedure

Changing the search engine also affects the cleaning procedures thus requiring an updated cleaning and filtering approach. To apply the new search strings to a Scopus search, to clean the data, and to initiate the study selection, the following procedure[4] needs to be applied:

1. Insert the search strings $S_1$–$S_8$ separately and use the time-range, i.e., conduct 8 individual searches for the required time slot of the update.
2. Set the automatic exclusion in Scopus using exclusion criterion $EC_2$ (Table 2) to:
   ``subject areas'' = computer science, engineering or multiple
3. Set the automatic exclusion in Scopus using exclusion criterion $EC_1$ (Table 2) to:
   ``language'' = ONLY English

[3]We used the word clouds to visually inspect the result set for "intruders," e.g., medicine, chemistry, and cancer therapy. Terms not matching our search criteria were collected and used to identify and remove the misselected papers from the result set.

[4]Please note: as our initial study resulted in a comprehensive Microsoft Excel spreadsheet, we also tailor the search and cleaning procedures to this tool. If you utilize a different tool, changes in the procedure might be necessary.

**Table 11  Final search strings used for the automatic database search in the study update procedure.**

| | Search string |
|---|---|
| $S_1$ | ((life-cycle **or** lifecycle **or** "life cycle") **and** (management **or** administration **or** development **or** description **or** authoring **or** deployment)) **and** (("software process" **and** ("software development model" **or** "process model")) **or** (SPI **or** "software process improvement")) |
| $S_2$ | (modeling **or** modelling **or** model-based **or** approach **or** variant) **and** (("software process" **and** ("software development model" **or** "process model")) **or** (SPI **or** "software process improvement")) |
| $S_3$ | (optimization **or** optimisation **or** customization **or** customisation **or** tailoring) **and** (("software process" **and** ("software development model" **or** "process model")) **or** (SPI **or** "software process improvement")) |
| $S_4$ | ("reference model" **or** "quality management" **or** evaluation **or** (assessment **or** audit) **or** (CMMI **or** "Capability Maturity Model Integration")) **and** (("software process" **and** ("software development model" **or** "process model")) **or** (SPI **or** "software process improvement")) |
| $S_5$ | ((feasibility **or** experience) **and** (study **or** report)) **and** (SPI **or** "software process improvement") |
| $S_6$ | ((life-cycle **or** lifecycle **or** "life cycle") **and** (design **or** modeling **or** modelling **or** analysis **or** training)) **and** (("software process" **and** ("software development model" **or** "process model")) **or** (SPI **or** "software process improvement")) |
| $S_7$ | (measurement **or** evaluation **or** approach **or** variant **or** improvement) **and** (("software process" **and** ("software development model" **or** "process model")) **or** (SPI **or** "software process improvement")) |
| $S_8$ | ((SCAMPI **or** "Standard CMMI Appraisal Method for Process Improvement") **or** (SPICE **or** "ISO/IEC 15504") **or** (PSP **or** "Personal Software Process") **or** (TSP **or** "Team Software Process")) **and** (("software process" **and** ("software development model" **or** "process model")) **or** (SPI **or** "software process improvement")) |

4. Export all search results into one Microsoft Excel file.
5. Eliminate duplicates ($EC_4$, Table 2) applying the *duplicate elimination function* in Microsoft Excel to the paper title (double-check and confirm by also checking authors and abstract).
6. Conduct the study selection procedures based on the inclusion and exclusion criteria listed in Table 2 following the procedure description in 'Analysis procedures'.

### Funding

Philipp Diebold's activities in this study were partially conducted in a Software Campus project funded by the German Ministry of Education and Research (BMBF 01IS12053). Marco Kuhrmann and Jürgen Münch received no funding for this work. The funders had no role in study design, data collection and analysis, decision to publish, or preparation of the manuscript.

### Grant Disclosures

The following grant information was disclosed by the authors:
German Ministry of Education and Research: BMBF 01IS12053.

### Competing Interests

The authors declare there are no competing interests.

## Author Contributions

- Marco Kuhrmann and Philipp Diebold conceived and designed the experiments, performed the experiments, analyzed the data, contributed reagents/materials/analysis tools, wrote the paper, prepared figures and/or tables.
- Jürgen Münch conceived and designed the experiments, performed the experiments, analyzed the data, contributed reagents/materials/analysis tools, reviewed drafts of the paper.

## Data Availability

The raw data has been supplied as Data S1.

## Supplemental Information

Supplemental information for this article can be found online at http://dx.doi.org/10.7717/peerj-cs.62#supplemental-information.

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
