# Peer review of "Software process improvement: a systematic mapping study on the state of the art"

_PeerJ Computer Science, doi:10.7717/peerj-cs.62_

## Round 0.1 · original submission · Minor Revisions

· Academic Editor

Minor Revisions

The reviewers are positive about the paper. It is clear that the systematic mapping study reported in this paper is thorough and conducted according to well accepted practices.

One of the reviewers requires a minor revision listing a few technical concerns and requests for clarification (see "Comments for the author" of the second review). The same reviewer notes some repetition between Discussion and Conclusion. The Conclusion section could perhaps be slightly shortened.

The reviewer also makes a comment about the overlap between this manuscript and the previous version of the paper. I leave it to the authors to decide if and how to handle this comment. It is clear that this manuscript substantially extends the conference paper and thus requirements are met in this respect. The conference papers has more authors than the ones listed in this submission - we assume that the authors that were left out from the conference paper are aware of it and agree.

The result set of the literature search is "raw data". In line with the review policy of PeerJ CS, the authors should submit this material for review as supplementary material. Please check point number 4 of the following instructions:
https://peerj.com/about/author-instructions/cs

It is claimed in the submission that the dataset is too large. In this case please upload it to an archived open data management system (PeerJ recommends using figshare).

I recommend the authors to package the result set as a table (e.g. Excel file) and upload it together with the revised version of the paper. I also recommend that the table with the result set should include a column classifying each paper according to the "publication objective" categorization introduced in lines 330-339 of page 11 (and any other additional metadata the authors deem useful). If the authors wish to submit the entire search result (on top of the result set) they are welcome to do so as well.

·

Basic reporting

The paper is excellently written, clear and unambiguous.

Experimental design

The research design is rigorous, conforming to all guidelines for systematic mapping. The method is extensively described, research questions clearly identified and subsequently addressed.

Validity of the findings

The findings are valid. More specifically the findings are useful to any researcher in the general area of software process improvement.

Additional comments

There have been so many poorly done systematic literature reviews that it was refreshing to find one that was done well, rigorously and produced valid and useful findings.

Reviewer 2 ·

Basic reporting

This paper is an extension of the authors' earlier paper providing a systematic mapping of Software Process Improvement updating their previous study findings.
The paper follows the guidelines of SLRs by Kitchenham and Charters as well as SMS by Petersen et al. The authors propose 3 research questions and describe their data collection and analysis procedures in great detail.

Experimental design

The motivation of the study is "to shed light on" the field of SPI and to present the state of the art of SPI. I would have preferred the motivation being related to findings or hypothesis of the previous studies by the authors.

Despite the fact that this paper is an extension of the authors' earlier mapping study of SPI, I find it seriously disturbing that the authors have identical abstract, introduction, related works section, research design, data collection and analysis sections to their earlier paper that was published at ICSSP conference in 2015. First, after having conducted such an extensive literature review, surely the authors can come up with a different abstract and introduction sections with citing different articles. Secondly, the extended paper could focus considerably more on the discussion if it wasn't repeating the first sections of the paper - a reference to the previous paper would have sufficed.

Validity of the findings

The authors describe their findings clearly even though it is a very technical and long section. The Discussion section is particularly valuable to this paper. At the same time, there is a lot of repetition between Discussion and Conclusion sections which the authors should revise.

Additional comments

There are some comments that the authors could take into account in improving their paper:
1. When a research topic is as applied as SPI and has been around for a long time, there could be little novelty in it for research to investigate much further (in Discussion/Limitation sections).
2. In Introduction, the authors say that "we still struggle to answer the question like: What is out there?..." Could you please refer to papers that would support such a claim - who is struggling?
3. About agile SPI, mentioned in Introduction, there should also be a reference to Salo and Abrahamsson whose agile SPI paper was one of the first ones and is heavily cited.
4. In Section 3.5 the authors describe how they called further researchers to confirm their publication classification. How did you select these researchers? How many were there? Did you have anybody from ISO/IEC JTC1 SC7 WG10 that develops ISO/IEC 15504/33000 standards? Both space and nuclear SPI fields are completely missing from the classification.
5. Table 11 - search strings. In the opinion of this reviewer, the authors have mixed up process models with methods, and assessment methods with improvement methods.
SPICE/ISO/IEC 15504 should have been in the S4 string rather than in S8 as it is, first and foremost, a process model (having both a process reference and a process assessment model in it) rather than a method. SPICE has a 7-step improvement cycle in ISO/IEC 15504 Part 7. CMMI improvement method is called IDEAL while SCAMPI is the assessment method for CMMI.

---

## Round 0.2 · accepted · Accept

· Academic Editor

Accept

This acceptance is conditional on you making the raw data publicly available (i.e. the table in the supplemental file 2016-04-19_data-final-cleaned.xlsx should be made available as "raw data")